# A Statistical Online Inference Approach in Averaged Stochastic Approximation

**Chuhan Xie**
School of Mathematical Sciences
Peking University
Beijing, 100871, China
ch_xie@pku.edu.cn

**Zhihua Zhang**
School of Mathematical Sciences
Peking University
Beijing, 100871, China
zhzhang@math.pku.edu.cn

## Abstract

In this paper we propose a general framework to perform statistical online inference in a class of constant step size stochastic approximation (SA) problems, including the well-known stochastic gradient descent (SGD) and Q-learning. Regarding a constant step size SA procedure as a time-homogeneous Markov chain, we establish a functional central limit theorem (FCLT) for it under weaker conditions, and then construct confidence intervals for parameters via random scaling. To leverage the FCLT results in the Markov chain setting, an alternative condition that is more applicable for SA problems is established. We conduct experiments to perform inference with both random scaling and other traditional inference methods, and finds that the former has a more accurate and robust performance.

## 1 Introduction

Stochastic approximation (SA) is a class of fixed-point algorithms that enjoys a wide range of applications and research [23, 24, 21, 14]. In general, with the goal of solving an underlying deterministic equation $z(\theta^*) = 0$, SA algorithms perform iterative updates based on random approximations of $z(\theta)$. An SA procedure has the following form

$$\theta_{t+1} = \theta_t - \eta Z_t, \quad t = 0, 1, 2, \dots, \tag{1}$$

where $\eta > 0$ is the constant step size and $Z_t = Z(\theta_t, \xi_{t+1})$ is a random realization satisfying $\mathbb{E}_{\xi_{t+1}} Z(\theta_t, \xi_{t+1}) = z(\theta_t)$. Although SA algorithms are initially proposed for solving optimization problems, a large variety of modern machine learning problems such as TD learning [25] and Q-learning [28] can also be solved via stochastic approximation, and hence studies of such problems in an SA perspective have emerged in recent years [6, 15, 20, 18].

The celebrated Polyak-Ruppert averaged estimator [22] is often used to stabilize and accelerate SA algorithms. Instead of using the last iterate of an SA procedure, the estimator takes an average over all iterates. It is well-known and extensively studied that if the step size $\eta$ in (1) is replaced by suitably decaying step sizes, the estimator converges almost surely to the ground-truth solution $\theta^*$, and a central limit theorem (CLT) with optimal covariance can be established for the averaged iterates. Examples include [5], which proposes the batch-means inference method for SGD-type estimator; and [4], which gives the very first asymptotic results and a statistical inference framework for adaptively collected data. Recently, this result has been extended to a functional central limit theorem (FCLT) in particular settings, such as SGD [16], Local SGD [17], and Q-learning [18]. In these works, however, additional conditions such as $(2 + \delta)$-th bounded moments of noises are required to obtain such stronger FCLT result, and proofs are based on an elaborated decomposition on the partial sum, which leads to some cumbersome calculation.

This motivates us to consider whether the same results can be established in a simpler way. In many real applications, constant step size SA algorithms are often used for convenience, and in this case

the SA algorithm can be seen as a time-homogeneous Markov chain because each iteration of the SA procedure is only based on the current state. Such perspective is also adopted by [9], which gives some nice theoretical results on the constant step size SGD. Meanwhile, FCLT results are well-established in the Markov chain setting [19, 11, 26, 3, 7, 8]. Due to the fine structure of such Markov chains, the FCLT may be established with weak conditions, i.e., with only second bounded moments of noises.

Based on the established FCLT, statistical inference via the random scaling method can be implemented. First proposed in [13] for time series regression and recently adapted in the inference of SGD parameters [16], random scaling utilizes the whole trajectory of the partial sum of iterates to construct a covariance-like matrix, and studentizes the averaged iterates using this matrix. This results in a pivotal statistic whose limit distribution does not depend on any other redundant parameter that needs to be further estimated. Without the need of the additional estimation step, so random scaling turns out to be more robust and efficient compared with traditional methods, such as spectral variance and batch-means [10].

The remainder of this paper is organized as follows. In Section 2 we propose a general statistical inference framework that includes the establishment of the FCLT and random scaling. In Section 3, we present applications of the framework in three modern problems including linear stochastic approximation (LSA), stochastic gradient descent (SGD) and Q-learning. Generally, the Polyak-Ruppert averaged estimator does not converge exactly to the ground-truth solution $\boldsymbol{\theta}^*$, but we give bounds on the bias to show such bias diminishes as the step size $\eta$ converges to zero. Numerical experiments are given in Section 4 to illustrate the accuracy and efficiency of our inference framework.

## 2 A Statistical Online Inference Framework

In this section, we introduce a general statistical inference framework for a class of constant step size stochastic approximation (SA) problems. The inference procedure contains three main steps: establishing a functional central limit theorem (FCLT) for the parameters to be estimated, constructing a pivotal statistic via random scaling based on the previously established FCLT, and providing a problem-dependent estimate on the bias between the ground-truth solution and the expectation of the stationary distribution to which the SA process converges.

### 2.1 Functional Central Limit Theorem

We first state a FCLT established under a Markov chain setting. Let $(X_n)_{n \geq 0}$ be an ergodic Markov chain on the measurable space $(S, \mathcal{S})$, with transition probability $P$ and stationary distribution $\pi$. For any measurable function $g \in L_\pi^2$, define the operator $Pg(x) = \int g(y)P(x, \mathrm{d}y) = \mathbb{E}[g(X_1)|X_0 = x]$. By $\mathbb{P}_x$ or $\mathbb{P}_\pi$ we denote the probability law of $(X_n)_{n \geq 0}$ starting from $x$ or having $\pi$ as its initial distribution; $\mathbb{E}_x$ or $\mathbb{E}_\pi$ the corresponding expectations. Given $f \in L_\pi^2$, we define the additive functionals $S_n(f) = \sum_{k=0}^{n-1} f(X_k)$ and $Y_n(t) = n^{-\frac{1}{2}} S_{\lfloor nt \rfloor}(f)$.

In [8], the authors proved a FCLT for a univariate functional of $(X_n)_{n \geq 0}$ started at a point $x$. With slight modification, we can obtain the following multivariate functional version.

**Theorem 2.1.** *Suppose* $\boldsymbol{f} \colon S \to \mathbb{R}^m$ *satisfies* $\int \|\boldsymbol{f}(x)\|^2 \pi(\mathrm{d}x) < \infty$ *and* $\int \boldsymbol{f}(x)\pi(\mathrm{d}x) = \boldsymbol{0}$. *If there exists* $0 < \alpha < \frac{1}{2}$ *such that*

$$\left\| \sum_{k=0}^{n-1} P^k \boldsymbol{f} \right\|_{L_\pi^2} = O(n^\alpha), \tag{2}$$

*then* $\boldsymbol{\Sigma_f} := \lim_{n \to \infty} \frac{1}{n} \mathbb{E}_\pi(S_n(\boldsymbol{f})S_n(\boldsymbol{f})^\top)$ *exists, and for* $\pi$-*almost every point* $x \in S$ *the sequence* $n^{-\frac{1}{2}} S_n(\boldsymbol{f})$ *converges in distribution, under the probability measure* $\mathbb{P}_x$, *to the* $d$-*dimensional Gaussian distribution* $\mathcal{N}(\boldsymbol{0}, \boldsymbol{\Sigma_f})$. *Furthermore, the process* $(Y_n(t))_{0 \leq t \leq 1}$ *converges weakly to* $(\boldsymbol{\Sigma_f}^{\frac{1}{2}} \boldsymbol{B}(t))_{0 \leq t \leq 1}$ *on the Skorokhod space* $D[0, 1]$, *where* $\boldsymbol{B} = (\boldsymbol{B}(t))_{t \geq 0}$ *is the* $d$-*dimensional standard Brownian motion.*

See Section A.1 for the detailed discussion and proof. In the sequel, we use $\xrightarrow{d}$ to denote "converge in distribution" and $\Rightarrow$ to denote "converge weakly in $D[0, 1]$". Note that an SA procedure of the form (1) can be seen as a time-homogeneous Markov chain, and therefore in order to establish a FCLT for

the parameter sequence $(\boldsymbol{\theta}_t)_{t\geq 0}$, it suffices to prove the existence of its stationary distribution $\pi$, and verify the conditions in Theorem 2.1 for $\boldsymbol{f} = \mathrm{Id} - \mathbb{E}_\pi \boldsymbol{\theta}$. The condition (2) is a bit harder to verify, so we attempt to ensure it with a stronger Wasserstein contraction condition. For $p, q \geq 1$, we define the Wasserstein distance $\mathcal{W}_{p,q}$ induced from the $L^p(\mathbb{R}^d)$ space by

$$\mathcal{W}_{p,q}(\mu, \nu) = \inf_{\gamma \in \Gamma(\mu,\nu)} \left( \int_{\mathbb{R}^d \times \mathbb{R}^d} \|x - y\|_p^q \mathrm{d}\gamma(x,y) \right)^{\frac{1}{q}},$$

where $\Gamma(\mu, \nu)$ is the set of all couplings of $\mu$ and $\nu$; and denote $\mathcal{P}_{p,q}$ as its corresponding Wasserstein space. We have the following proposition.

**Proposition 2.1.** *Suppose for some $p, q \geq 1$, there exists a constant $\gamma \in (0, 1)$ such that for any $\mu, \nu \in \mathcal{P}_{p,q}$, the Wasserstein contraction*

$$\mathcal{W}_{p,q}(\mu P, \nu P) \leq \gamma \mathcal{W}_{p,q}(\mu, \nu) \tag{3}$$

*holds. Suppose in addition that $\mathcal{W}_{p,q}(\delta(\cdot), \pi) \in L_\pi^2$, where $\delta(\cdot)$ is the Dirac measure and $\pi$ is the stationary distribution. Then for any $\boldsymbol{f} \colon \mathbb{R}^d \to \mathbb{R}^m$ satisfying $\int \|\boldsymbol{f}(x)\|^2 \pi(\mathrm{d}x) < \infty$, $\int \boldsymbol{f}(x)\pi(\mathrm{d}x) = \boldsymbol{0}$ and*

$$\|\boldsymbol{f}(\boldsymbol{x}) - \boldsymbol{f}(\boldsymbol{y})\|_\infty \leq \mathrm{Lip}_p(\boldsymbol{f})\|\boldsymbol{x} - \boldsymbol{y}\|_p \quad \text{for all } \boldsymbol{x}, \boldsymbol{y} \in \mathbb{R}^d,$$

*for some constant $\mathrm{Lip}_p(\boldsymbol{f})$ (i.e., each component of $\boldsymbol{f}$ is $\mathrm{Lip}_p(\boldsymbol{f})$-Lipschitz in the $\|\cdot\|_p$ norm), the condition (2) holds.*

The proof of Proposition 2.1 is deferred to Section A.2. As we will see later, many SA problems (e.g., linear SA, SGD and Q-learning) possess such a Wasserstein contraction property, and hence the corresponding FCLT holds.

## 2.2 Random Scaling

Once the FCLT for an SA problem is established, we can use the random scaling method to carry out inference. For the sequence $(\boldsymbol{\theta}_t)_{t\geq 0}$ which is supposed to converge to its stationary distribution $\pi$ and satisfy the FCLT

$$\frac{1}{\sqrt{n}} \sum_{k=1}^{\lfloor nt \rfloor} (\boldsymbol{\theta}_k - \mathbb{E}_\pi \boldsymbol{\theta}) \Rightarrow \boldsymbol{\Sigma}^{\frac{1}{2}} \boldsymbol{B}(t), \quad 0 \leq t \leq 1, \tag{4}$$

we studentize it with a matrix $\widehat{\boldsymbol{V}}_n$ constructed as follows

$$\widehat{\boldsymbol{V}}_n := \frac{1}{n} \sum_{j=1}^{n} \left\{ \frac{1}{\sqrt{n}} \sum_{i=1}^{j} (\boldsymbol{\theta}_i - \bar{\boldsymbol{\theta}}_n) \right\} \left\{ \frac{1}{\sqrt{n}} \sum_{i=1}^{j} (\boldsymbol{\theta}_i - \bar{\boldsymbol{\theta}}_n) \right\}^\top, \tag{5}$$

where $\bar{\boldsymbol{\theta}}_n := \frac{1}{n} \sum_{k=1}^{n} \boldsymbol{\theta}_k$ is the Polyak-Ruppert averaged estimator for the parameter $\boldsymbol{\theta}$. Such studentization leads to a pivotal statistic whose asymptotic distribution does not depend on the parameters of the problem.

**Proposition 2.2.** *Suppose the sequence $(\boldsymbol{\theta}_t)_{t\geq 0}$ converges to its stationary distribution $\pi$ and its FCLT (4) holds. For a matrix $\boldsymbol{R} \in \mathbb{R}^{\ell \times d}$ of rank $\ell \leq d$, we have*

$$n \left( \boldsymbol{R}(\bar{\boldsymbol{\theta}}_n - \mathbb{E}_\pi \boldsymbol{\theta}) \right)^\top \left( \boldsymbol{R}\widehat{\boldsymbol{V}}_n \boldsymbol{R}^\top \right)^{-1} \left( \boldsymbol{R}(\bar{\boldsymbol{\theta}}_n - \mathbb{E}_\pi \boldsymbol{\theta}) \right)$$

$$\xrightarrow{d} \boldsymbol{B}_\ell(1)^\top \left( \int_0^1 (\boldsymbol{B}_\ell(r) - r\boldsymbol{B}_\ell(1))(\boldsymbol{B}_\ell(r) - r\boldsymbol{B}_\ell(1))^\top \mathrm{d}r \right)^{-1} \boldsymbol{B}_\ell(1),$$

*where $\boldsymbol{B}_\ell$ is a $\ell$-dimensional standard Brownian motion and $\widehat{\boldsymbol{V}}_n$ is defined as in (5).*

Proposition 2.2 follows directly from the continuous mapping theorem. Statistical inference for $\mathbb{E}_\pi \boldsymbol{\theta}$ can be carried out via Proposition 2.2. For example, by setting $\boldsymbol{R} = \boldsymbol{I}_d$ we can construct a $(1-\alpha)$ asymptotic ellipsoidal confidence region centered at $\bar{\boldsymbol{\theta}}_n$ for estimating the whole vector $\mathbb{E}_\pi \boldsymbol{\theta}$. More often, we set $\boldsymbol{R} = \boldsymbol{e}_j$ (the $j$-th unit vector) to construct a $(1-\alpha)$ asymptotic confidence interval for estimating the $j$-th component of the expectation $(\mathbb{E}_\pi \boldsymbol{\theta})_j$, so as to avoid heavy computation from the matrix inverse $(\boldsymbol{R}\widehat{\boldsymbol{V}}_n \boldsymbol{R}^\top)^{-1}$.

**Corollary 2.1.** *Under the same assumptions of Proposition 2.2, we have that*

$$\mathbb{P}\left( (\bar{\boldsymbol{\theta}}_n)_j - q_{\frac{\alpha}{2}}\sqrt{\widehat{V}_{n,jj}/n} \le (\mathbb{E}_\pi \boldsymbol{\theta})_j \le (\bar{\boldsymbol{\theta}}_n)_j + q_{\frac{\alpha}{2}}\sqrt{\widehat{V}_{n,jj}/n} \right) \to 1 - \alpha,$$

*where $\widehat{V}_{n,jj}$ is the $j$-th diagonal element of $\widehat{V}_n$, and $q_{\frac{\alpha}{2}}$ is the $(1 - \alpha/2)$-th quantile of the following random variable*

$$t^* := \left( \int_0^1 (B_1(r) - rB_1(1))^2 \mathrm{d}r \right)^{-\frac{1}{2}} B_1(1), \tag{6}$$

*with $(B_1(t))_{t \ge 0}$ a one-dimensional standard Brownian motion.*

Table 1: Asymptotic critical values $q_\alpha$ defined by $q_\alpha = \min\{t : \mathbb{P}(t^* \le t) \ge 1 - \alpha\}$.

| $(1 - \alpha)$ | 1% | 2.5% | 5% | 10% | 50% | 90% | 95% | 97.5% | 99% |
|---|---|---|---|---|---|---|---|---|---|
| $q_\alpha$ | -8.634 | -6.753 | -5.324 | -3.877 | 0.000 | 3.877 | 5.324 | 6.753 | 8.634 |

The random variable $t^*$ is mixed normal with its distribution symmetric around zero. For easy reference, its critical values are listed in Table 1.[1]

It is crucial to explain why inference via random scaling takes advantage over traditional inference procedures (e.g., the spectral variance method and the batch-means method that will be introduced hereafter) in our SA problem. Traditional estimators are mostly constructed on top of the CLT

$$\frac{1}{\sqrt{n}} \sum_{k=1}^n (\boldsymbol{\theta}_k - \mathbb{E}_\pi \boldsymbol{\theta}) \xrightarrow{d} \mathcal{N}(\mathbf{0}, \boldsymbol{\Sigma}), \tag{7}$$

where $\boldsymbol{\Sigma} = \lim_{n \to \infty} \frac{1}{n} \mathbb{E}_\pi \left[ \left( \sum_{k=1}^n (\boldsymbol{\theta}_k - \mathbb{E}_\pi \boldsymbol{\theta}) \right) \left( \sum_{k=1}^n (\boldsymbol{\theta}_k - \mathbb{E}_\pi \boldsymbol{\theta}) \right)^\top \right]$ is the asymptotic covariance. Because $\boldsymbol{\Sigma}$ is unknown, one attempts to first construct an estimate $\widehat{\boldsymbol{\Sigma}}$ for $\boldsymbol{\Sigma}$ based on parameter samples $(\boldsymbol{\theta}_t)_{t \ge 0}$ and then establish a confidence interval based on (7) with $\boldsymbol{\Sigma}$ replaced by $\widehat{\boldsymbol{\Sigma}}$. As a consequence, the estimation error in $\widehat{\boldsymbol{\Sigma}}$ may cause significant impact on the inference efficiency. Meanwhile, during the estimation of $\boldsymbol{\Sigma}$, additional hyper-parameters are likely to occur, and different choices of such hyper-parameters may affect the inference performance. Random scaling, on the other hand, studentize the sum $\frac{1}{\sqrt{n}} \sum_{k=1}^n (\boldsymbol{\theta}_k - \mathbb{E}_\pi \boldsymbol{\theta})$ to obtain a pivotal statistic. It does not need the additional estimation step and additional hyper-parameters, resulting in its higher accuracy over traditional estimators.

### 2.3 Problem-Dependent Bias

The random scaling method mentioned above has already provided an efficient way to carry out inference on $(\mathbb{E}_\pi \boldsymbol{\theta})_j$ (or more generally, any linear combination of the expectation $\mathbb{E}_\pi \boldsymbol{\theta}$). However, in most nonlinear SA problems there is often a gap between the expectation $\mathbb{E}_\pi \boldsymbol{\theta}$ and the ground-truth solution $\boldsymbol{\theta}^*$. This issue will be further illustrated by the examples discussed in Section 3. Therefore, it is theoretically not valid to carry out inference on $\boldsymbol{\theta}^*$ with a confidence interval constructed following Corollary 2.1. However, in the following applications we try to upper bound the bias $\|\mathbb{E}_\pi \boldsymbol{\theta} - \boldsymbol{\theta}^*\|$ with a quantity of order $O(\eta^\alpha)$ for some $\alpha > 0$. When the step size is small, such bias is negligible and thus the original inference procedure may still be reasonable.

## 3 Case Studies

In Section 2, we have established a general statistical inference framework for constant step size stochastic approximation problems. We now illustrate several specific applications that are well-known in optimization and reinforcement learning.

---

[1] These critical values are quoted from [17] (Table 2, the $\beta = 0$ case). In that paper the authors used kernel density estimation to smooth the empirical density function before obtaining critical values. Critical values without smoothing data in advance can be found in [1, 13].

## 3.1 Linear Stochastic Approximation (LSA)

The first example considers solving a linear system of the form $A\theta = b$, where $A \in \mathbb{R}^{d \times d}$ and $b \in \mathbb{R}^d$. We assume that $A$ is invertible, and there is a unique solution $\theta^*$ to the linear equation. Suppose we can observe a sequence of random variables of the form $\{(A_t, b_t)\}_{t \geq 1}$, assumed to be independent and identically distributed, and unbiased in the sense that $\mathbb{E}(A_t | \mathcal{F}_{t-1}) = A$ and $\mathbb{E}(b_t | \mathcal{F}_{t-1}) = b$, where $\mathcal{F}_{t-1}$ denotes the $\sigma$-field generated by $\{(A_k, b_k)\}_{k=1}^{t-1}$. For a given initial vector $\theta_0$, LSA is formulated as

$$\theta_{t+1} = \theta_t - \eta(A_{t+1}\theta_t - b_{t+1}), \tag{8}$$

where $\eta > 0$ is the constant step size. Before presenting our results, we make some assumptions.

**Assumption 3.1.** *The matrix $-A \in \mathbb{R}^{d \times d}$ is Hurwitz, i.e., $\lambda := \min_{1 \leq i \leq d} \Re(\lambda_i(A)) > 0$ where $\lambda_i(A)$ is the $i$-th eigenvalue of $A$ and $\Re(\lambda_i(A))$ is its corresponding real part.*

**Assumption 3.2** (Bounded second moment of noise). *Denote $\Xi_t = A_t - A$ and $\xi_t = b_t - b$. There exist $v_A^2 > 0$ and $v_b^2 > 0$ such that $\mathbb{E}\|\Xi_t u\|^2 \leq v_A^2$ and $\mathbb{E}|\xi_t^\top u|^2 \leq v_b^2$ for any $t \geq 1$ and $u \in \mathbb{S}^{d-1}$. Moreover, $\Xi_t$ and $\xi_t$ are uncorrelated.*

**Theorem 3.1.** *Suppose Assumptions 3.1 and 3.2 hold for the LSA procedure* (8). *Then there exists a constant $\eta_0 > 0$ such that for any $\eta \in (0, \eta_0)$,*

1. *the Markov process $(\theta_t)_{t \geq 0}$ defined by* (8) *has a unique stationary distribution $\pi_\eta$;*

2. *for $\theta \sim \pi_\eta$, we have $\mathbb{E}_{\pi_\eta}\theta = \theta^*$;*

3. *the following FCLT holds*

$$\frac{1}{\sqrt{n}} \sum_{k=1}^{\lfloor nt \rfloor} (\theta_k - \theta^*) \Rightarrow \Sigma_{\pi_\eta}^{\frac{1}{2}} B(t), \quad 0 \leq t \leq 1,$$

*where $(B(t))_{t \geq 0}$ is the $d$-dimensional standard Brownian motion and $\Sigma_{\pi_\eta} = \lim_{n \to \infty} n\mathrm{cov}_{\pi_\eta}(\bar{\theta}_n)$ is the asymptotic covariance matrix.*

**Remark 3.1.** *If we further suppose that $\{A_t\}_{t \geq 1}$ and $\{b_t\}_{t \geq 1}$ have i.i.d. entries, then the asymptotic covariance matrix $\Sigma_{\pi_\eta}$ has an explicit form (see Section A.3 for details). This matches the asymptotic results in [20], where the same assumptions are needed and a central limit theorem (CLT) is established, based on an additional requirement that the noises $\Xi_t$ and $\xi_t$ have bounded $(2 + \delta)$-th moments for some $\delta > 0$. Theorem 3.1 extends this CLT result into a functional version even without such an additional moment requirement.*

Note that in the LSA problem, the expectation of the stationary distribution $\mathbb{E}_{\pi_\eta}\theta$ exactly matches the unique solution $\theta^*$, so that the problem-dependent bias in this case vanishes. The established FCLT allows us to perform statistical inference directly on $\theta^*$ via Corollary 2.1.

## 3.2 Stochastic Gradient Descent (SGD)

Next, we consider the inference problem for a minimizer of an objective function, define by

$$\theta^* := \underset{\theta \in \mathbb{R}^d}{\arg\min}\, f(\theta),$$

where $f(\theta) := \mathbb{E}[f(\theta, \xi)]$ and $f(\theta, \xi)$ is an unbiased random variable for the objective function. The constant step size SGD iteratively updates the parameter in the following form

$$\theta_{t+1} = \theta_t - \eta \nabla f(\theta_t, \xi_{t+1}). \tag{9}$$

In this case, we make the following assumptions to derive our results.

**Assumption 3.3.** *$f(\cdot)$ is $m$-strongly convex, and for each $\xi$, $f(\cdot, \xi)$ is $L$-smooth.*

**Assumption 3.4** (Regularized gradient noise). *Define $\varepsilon(\theta, \xi) = \nabla f(\theta, \xi) - \nabla f(\theta)$ as the gradient noise at $\theta$, and $S = \mathbb{E}[\varepsilon(\theta^*, \xi)\varepsilon(\theta^*, \xi)^\top]$. There exists some $C > 0$ such that*

$$\left\|\mathbb{E}[\varepsilon(\theta, \xi)\varepsilon(\theta, \xi)^\top] - S\right\| \leq C\left[\|\theta - \theta^*\| + \|\theta - \theta^*\|^2\right]. \tag{10}$$

**Theorem 3.2.** *Suppose Assumptions 3.3 and 3.4 hold for the SGD iterates* (9). *Then there exists a constant $\eta_0 > 0$ such that for any $\eta \in (0, \eta_0)$,*

1. *the Markov process $(\boldsymbol{\theta}_t)_{t \geq 0}$ defined by* (9) *has a unique stationary distribution $\pi_\eta$;*

2. *for $\boldsymbol{\theta} \sim \pi_\eta$, we have $\mathbb{E}_{\pi_\eta} \|\boldsymbol{\theta} - \boldsymbol{\theta}^*\|^2 \leq \frac{\eta}{m} \mathbb{E}_{\pi_\eta} \|\boldsymbol{\varepsilon}(\boldsymbol{\theta}, \xi)\|^2 = O(\eta)$;*

3. *The following FCLT holds*

$$\frac{1}{\sqrt{n}} \sum_{k=1}^{\lfloor nt \rfloor} (\boldsymbol{\theta}_k - \mathbb{E}_{\pi_\eta} \boldsymbol{\theta}) \Rightarrow \boldsymbol{\Sigma}_{\pi_\eta}^{\frac{1}{2}} \boldsymbol{B}(t), \quad 0 \leq t \leq 1, \tag{11}$$

*where $(\boldsymbol{B}(t))_{t \geq 0}$ is the $d$-dimensional standard Brownian motion and $\boldsymbol{\Sigma}_{\pi_\eta} = \lim_{n \to \infty} n\mathrm{cov}_{\pi_\eta}(\bar{\boldsymbol{\theta}}_n)$ is the asymptotic covariance matrix.*

Theorem 3.2 provides all ingredients for performing inference on $\mathbb{E}_{\pi_\eta} \boldsymbol{\theta}$. In addition, the bias can be bounded via

$$\|\mathbb{E}_{\pi_\eta} \boldsymbol{\theta} - \boldsymbol{\theta}^*\| \leq \left(\mathbb{E}_{\pi_\eta} \|\boldsymbol{\theta} - \boldsymbol{\theta}^*\|^2\right)^{\frac{1}{2}} \leq \sqrt{\frac{\eta}{m}} \left(\mathbb{E}_{\pi_\eta} \|\boldsymbol{\varepsilon}(\boldsymbol{\theta}, \xi)\|^2\right)^{\frac{1}{2}} \leq \sqrt{\frac{\eta}{m}} \left(\mathbb{E}_{\pi_\eta} \|\nabla f(\boldsymbol{\theta}, \xi)\|^2\right)^{\frac{1}{2}},$$

that is, it is of order $O(\eta^{1/2})$.

### 3.3 Q-Learning

Finally we consider an application in Q-learning. An infinite-horizon MDP is represented by $\mathcal{M} = (\mathcal{S}, \mathcal{A}, \gamma, P, R, r)$, where $\mathcal{S}$ is the state space and $\mathcal{A}$ is the action space, $\gamma \in (0, 1)$ is the discount factor, $P \colon \mathcal{S} \times \mathcal{A} \to \Delta(\mathcal{S})$ represents the probability transition kernel, $R \colon \mathcal{S} \times \mathcal{A} \to [0, \infty)$ is the random reward, and $r \colon \mathcal{S} \times \mathcal{A} \to [0, \infty)$ is the expectation of the reward. For a given deterministic policy $\pi \colon \mathcal{S} \to \mathcal{A}$, the expected long-term reward is measured by a value function $V^\pi$ and a Q-function $Q^\pi$ defined as follows:

$$V^\pi(s) = \mathbb{E}_{\tau \sim \pi} \left[ \sum_{t=0}^{\infty} \gamma^t r(s_t, a_t) \Big| s_0 = s \right] \text{ and } Q^\pi(s, a) = \mathbb{E}_{\tau \sim \pi} \left[ \sum_{t=0}^{\infty} \gamma^t r(s_t, a_t) \Big| s_0 = s, a_0 = a \right],$$

for any state-action pair $(s, a) \in \mathcal{S} \times \mathcal{A}$. Here $\tau = \{(s_t, a_t)\}_{t \geq 0}$ is a trajectory of the MDP induced by the policy $\pi$ and the expectation $\mathbb{E}_{\tau \sim \pi}(\cdot)$ is taken with respect to the randomness of the trajectory $\tau$. The optimal value function $V^*$ and optimal Q-function $Q^*$ are defined as $V^*(s) = \max_\pi V^\pi(s)$ and $Q^*(s, a) = \max_\pi Q^\pi(s, a)$, respectively. It is well known that $Q^*$ is the unique fixed point of the Bellman operator $\mathcal{T}$, i.e., $\mathcal{T}(Q^*) = Q^*$, where

$$\mathcal{T}(Q)(s, a) = r(s, a) + \gamma \mathbb{E}_{s' \sim P(\cdot|s,a)} \max_{a' \in \mathcal{A}} Q(s', a').$$

Now assume access to a generative model. In particular, in each iteration $t \geq 0$, we collect independent samples of the reward $r_t(s, a)$ and the next state $s_t(s, a) \sim P(\cdot|s, a)$ for every state-action pair $(s, a) \in \mathcal{S} \times \mathcal{A}$. The constant step size Q-learning maintains a Q-function estimate, $Q_t \colon \mathcal{S} \times \mathcal{A} \to \mathbb{R}$, for all $t \geq 0$ and updates its entries by the following update rule

$$Q_{t+1} = (1 - \eta)Q_t + \eta \widehat{\mathcal{T}}_{t+1}(Q_t), \tag{12}$$

where $\eta \in (0, 1]$ is the step size and $\widehat{\mathcal{T}}_t$ is the empirical Bellman operator constructed by samples collected in the $t$-th iteration:

$$\widehat{\mathcal{T}}_t(Q)(s, a) = r_t(s, a) + \gamma \max_{a' \in \mathcal{A}} Q(s_t, a'), \text{ where } r_t(s, a) \sim R(s, a) \text{ and } s_t = s_t(s, a) \sim P(\cdot|s, a). \tag{13}$$

Clearly, $\widehat{\mathcal{T}}_t$ is an unbiased estimate of the Bellman operator $\mathcal{T}$.

**Assumption 3.5** (Uniformly bounded reward). *The random reward $R$ is non-negative and uniformly bounded, i.e., $0 \leq R(s, a) \leq 1$ almost surely for all $(s, a) \in \mathcal{S} \times \mathcal{A}$.*

**Theorem 3.3.** *Suppose Assumption 3.5 holds for the Q-learning update rule* (12), *and assume access to a generative model for each state-action pair $(s, a) \in \mathcal{S} \times \mathcal{A}$. Let $\boldsymbol{Q} \in \mathbb{R}^{|\mathcal{S}| \times |\mathcal{A}|}$ be the matrix form of the Q-function $Q(s, a)$. Then there exists a constant $\eta_0 > 0$ such that for any $\eta \in (0, \eta_0)$,*

1. *the Markov process* $(\boldsymbol{Q}_t)_{t \geq 0}$ *defined by* (12) *has a unique stationary distribution* $\mathcal{Q}_\eta$;

2. *for* $\boldsymbol{Q} \sim \mathcal{Q}_\eta$, *we have* $\mathbb{E}_{\mathcal{Q}_\eta} \| \boldsymbol{Q} - \boldsymbol{Q}^* \|_\infty = O(\eta^{1/2})$;

3. *The following FCLT holds*

$$\frac{1}{\sqrt{n}} \sum_{k=1}^{\lfloor nt \rfloor} (\boldsymbol{Q}_k - \mathbb{E}_{\mathcal{Q}_\eta} \boldsymbol{Q}) \Rightarrow \boldsymbol{\Sigma}_{\mathcal{Q}_\eta}^{\frac{1}{2}} \boldsymbol{B}(t), \quad 0 \leq t \leq 1, \tag{14}$$

*where* $(\boldsymbol{B}(t))_{t \geq 0}$ *is the* $d$-*dimensional standard Brownian motion and* $\boldsymbol{\Sigma}_{\mathcal{Q}_\eta} = \lim_{n \to \infty} n \mathrm{cov}_{\mathcal{Q}_\eta}(\bar{\boldsymbol{Q}}_n)$ *is the asymptotic covariance matrix.*

## 4 Experiments

In this section we conduct the empirical analysis of the random scaling method via Monte Carlo experiments. Two traditional inference methods, namely, the spectral variance method and the batch-means method, are also examined for comparison. These two methods are popular in estimating the asymptotic variance of a Markov chain [10], and confidence intervals can be constructed based on the CLT (7) with $\boldsymbol{\Sigma}$ replaced by an estimator $\widehat{\boldsymbol{\Sigma}}$. From now on, we represent the concerned parameter as $\boldsymbol{\beta}$ for clarity.

The spectral variance estimator is based on the fact that $\boldsymbol{\Sigma} = \sum_{s=-\infty}^{+\infty} \mathrm{cov}_{\pi_\eta}(\boldsymbol{\beta}_t, \boldsymbol{\beta}_{t+s})$. It is constructed as

$$\widehat{\boldsymbol{\Sigma}}_{SV} := \sum_{s=-(b_n-1)}^{b_n-1} w_n(s) \gamma_n(s),$$

where $w_n(\cdot)$ is the lag window, $b_n$ is the truncation point, and

$$\gamma_n(s) = \gamma_n(-s)^\top := \frac{1}{n} \sum_{i=1}^{n-s} (\boldsymbol{\beta}_i - \bar{\boldsymbol{\beta}}_n)(\boldsymbol{\beta}_{i+s} - \bar{\boldsymbol{\beta}}_n)^\top, \quad s \geq 0,$$

which is a consistent estimator of the lag $s$ autocovariance $\gamma(s) := \mathrm{cov}_{\pi_\eta}(\boldsymbol{\beta}_t, \boldsymbol{\beta}_{t+s})$. Under mild conditions on $w_n(\cdot)$ and $b_n$, $\widehat{\boldsymbol{\Sigma}}_{SV}$ is a consistent estimator of $\boldsymbol{\Sigma}$ [10]. In our experiments we choose $w_n(\cdot)$ to be the Tukey-Hanning window $w_n(k) = \left(\frac{1}{2} + \frac{1}{2} \cos\left(\pi |k|/b_n\right)\right) \mathbf{1}_{\{|k| < b_n\}}$ with $b_n = n^{3/4}$.

The batch-means estimator is another consistent estimator of $\boldsymbol{\Sigma}$. It first divides a total of $n = a_n b_n$ iterates into $a_n$ batches, and calculate the mean of each batch $\bar{\boldsymbol{\beta}}_{k:k+1} := \frac{1}{b_n} \sum_{i=1}^{b_n} \boldsymbol{\beta}_{kb_n+i}$ for $k = 0, \ldots, a_n - 1$. The batch-means estimator is then defined as

$$\widehat{\boldsymbol{\Sigma}}_{BM} := \frac{b_n}{a_n - 1} \sum_{k=0}^{a_n-1} (\bar{\boldsymbol{\beta}}_{k:k+1} - \bar{\boldsymbol{\beta}}_n)(\bar{\boldsymbol{\beta}}_{k:k+1} - \bar{\boldsymbol{\beta}}_n)^\top.$$

When $a_n$ and $b_n$ are allowed to increase as $n$ increases (e.g., by setting $a_n = b_n = \lfloor \sqrt{n} \rfloor$), $\widehat{\boldsymbol{\Sigma}}_{BM}$ is a consistent estimator of $\widehat{\boldsymbol{\Sigma}}$ [12]. In our experiments we choose $a_n = b_n = \lfloor \sqrt{n} \rfloor$.

In the following, we consider the constant step size SGD on two baseline models: linear regression and logistic regression. Additional experiments for Q-learning on Grid World are included in Appendix D. All experiments are run on Intel Xeon 14-core CPUs. The performance of the spectral variance method, batch-means method, and random scaling method are compared in three aspects: the coverage rates for $\mathbb{E}_{\pi_\eta} \boldsymbol{\beta}$ and $\boldsymbol{\beta}^*$, and the lengths of confidence intervals. The nominal coverage rate is chosen as 95%. For brevity, we focus on the first coefficient $\boldsymbol{\beta}_1$ hereafter. The implementation code is available in `https://github.com/bangoz/sa-inference`.

**Linear Regression**  The data are generated from

$$y_t = \boldsymbol{x}_t^\top \boldsymbol{\beta}^* + \varepsilon_t, \quad t \geq 1, \tag{15}$$

where $\boldsymbol{x}_t$ is a $d$-dimensional covariate following the multivariate normal distribution $\mathcal{N}(0, I_d)$, $\varepsilon_t$ is the noise from $\mathcal{N}(0, 1)$, and $\boldsymbol{\beta}^*$ is equi-spaced on the interval $[0, 1]$. The dimension of $\boldsymbol{x}$ is set as

$d = 5, 20$. The constant learning rate is set as $\eta = 0.01, 0.05, 0.1$. The initial value of $\beta_0$ is set as zero. The simulation results are based on 1,000 replications. Our experimental design is the same as that of [29, 16], except that our learning rates $\eta$ are smaller than the initial learning rates in their settings, which are chosen to be $\eta_0 = 0.5, 1$. This is inevitable because constant step size SGD requires smaller learning rates to ensure convergence. Note that SGD for linear regression is also a case of LSA, so the expectation of the stationary distribution is equal to the true parameter, i.e., $\mathbb{E}_{\pi_\eta} \boldsymbol{\beta} = \boldsymbol{\beta}^*$.

Table 2 summarizes partial simulation results for $d = 5$. Full results for $d = 5, 20$ can be found in Appendix C. Overall, the random scaling method is satisfactory in that its coverage rates are the closest to the nominal one (95%) across all experimental settings. Nevertheless, the average lengths of confidence intervals (CI) constructed via random scaling are always larger than that of both the spectral variance method and the batch-means method. The batch-means method, on the contrary, shows the lowest coverage rates and the smallest average CI lengths, and its performance varies hugely when we choose different step sizes. When $\eta = 0.01$ it performs the worst, deviating from the nominal rate about 10% even at $n = 40,000$ (when $d = 5$) or at $n = 100,000$ (when $d = 20$). Comparing our experiment results with that of [16], where the authors use SGD with decaying step sizes of the form $\eta_t = \eta_0 t^{-a}$, we find that average CI lengths in our settings are a bit larger. Such observation is also reasonable, because SGD with particular decaying step sizes is shown to converge to the true parameter almost surely, and so does its corresponding Polyak-Ruppert averaged estimator [22]; but in the case of constant step sizes, the iterates converge to a non-degenerate stationary distribution, and such non-degeneration adds to additional fluctuation of the Polyak-Ruppert averaged estimator.

**Logistic Regression**    The data are generated from

$$\mathbb{P}(y_t = 1) = \frac{1}{1 + \exp(-\boldsymbol{x}_t^\top \boldsymbol{\beta}^*)}, \quad \mathbb{P}(y_t = 0) = \frac{\exp(-\boldsymbol{x}_t^\top \boldsymbol{\beta}^*)}{1 + \exp(-\boldsymbol{x}_t^\top \boldsymbol{\beta}^*)}, \quad t \geq 1. \tag{16}$$

All the settings are the same as in linear regression. Note that for logistic regression, the expectation of the stationary distribution is not necessarily equal to the true parameter, i.e., $\mathbb{E}_{\pi_\eta} \boldsymbol{\beta} \neq \boldsymbol{\beta}^*$. However, since the difference is diminishing as $\eta \to 0$, the confidence intervals may still be useful for inference on $\boldsymbol{\beta}^*$. Therefore, we include the coverage rates for $\boldsymbol{\beta}_1^*$ in our results.

Table 3 summarizes partial simulation results for $d = 5$. Full results can be found in Appendix C. These results are similar to those in linear regression. Coverage rates for $\boldsymbol{\beta}_1^*$ are generally smaller than coverage rates for $\mathbb{E}_{\pi_\eta} \boldsymbol{\beta}_1^*$ as expected, and their differences become larger as the step size gets larger. Nevertheless, coverage rates for $\boldsymbol{\beta}_1^*$ based on random scaling deviates within 5% of the nominal one in most settings, which indicates that such confidence intervals are still reasonable when the step size is small. The batch-means method is still not robust to different step sizes, because the coverage rates even fall below 60% when $\eta = 0.01$.

## 5    Conclusion

In this paper we have studied statistical inference on a class of constant step size stochastic approximation (SA) problems. Based on a FCLT in the context of Markov chains, we have established the corresponding FCLTs for LSA, SGD, and Q-learning. A sufficient condition for the FCLT has been stated and proved for ease of application. Various inference methods including the random scaling method, spectral variance method and batch-means method have been conducted in experiments, based on the established FCLTs. We have shown that random scaling performs well in terms of coverage rates, and is also robust against different step sizes.

There are also limitations and issues that need to be further investigated. First, except in trivial cases, iterates converge to a stationary distribution whose expectation does not equal to the true parameter, and such systematic bias prohibits efficient inference procedures for the true parameter we are concerned about. It remains open whether there exist practical ways to mitigate the impact of such bias. Probabilistic tools such as the large deviation theory might help construct valid confidence intervals for the true parameter. Second, recent works considering SA problems with decaying step sizes, such as [16, 17, 18], prove the FCLT by manually dividing the estimator into several terms, applying martingale CLT to one term, and uniformly bounding other terms. It is

Table 2: Linear Regression, $d = 5$

|  |  | $n = 5,000$ | $n = 10,000$ | $n = 20,000$ | $n = 40,000$ |
|---|---|---|---|---|---|
| $\eta = 0.01$ |  |  |  |  |  |
|  | Spectral Variance | 88.6(1.005) | 91.1(0.900) | 91.5(0.881) | 92.4(0.837) |
| Cov. for $\boldsymbol{\beta}_1^*$ (%) | Batch Means | 69.8(1.451) | 77.8(1.314) | 82.3(1.206) | 86.7(1.073) |
|  | Random Scaling | 92.8(0.817) | 94.6(0.714) | 95.0(0.689) | 94.9(0.695) |
|  | Spectral Variance | 0.050(0.011) | 0.036(0.007) | 0.026(0.004) | 0.018(0.003) |
| Length | Batch Means | 0.030(0.003) | 0.024(0.002) | 0.019(0.001) | 0.015(0.001) |
|  | Random Scaling | 0.074(0.033) | 0.053(0.022) | 0.036(0.014) | 0.026(0.010) |
| $\eta = 0.05$ |  |  |  |  |  |
|  | Spectral Variance | 91.9(0.862) | 91.6(0.877) | 90.8(0.913) | 91.6(0.877) |
| Cov. for $\boldsymbol{\beta}_1^*$ (%) | Batch Means | 90.0(0.948) | 93.6(0.773) | 92.7(0.822) | 92.0(0.857) |
|  | Random Scaling | 94.9(0.695) | 95.2(0.675) | 95.4(0.662) | 94.1(0.745) |
|  | Spectral Variance | 0.055(0.012) | 0.039(0.007) | 0.028(0.005) | 0.020(0.003) |
| Length | Batch Means | 0.050(0.004) | 0.038(0.002) | 0.027(0.001) | 0.020(0.001) |
|  | Random Scaling | 0.076(0.031) | 0.055(0.023) | 0.039(0.015) | 0.027(0.010) |
| $\eta = 0.1$ |  |  |  |  |  |
|  | Spectral Variance | 89.5(0.969) | 89.8(0.957) | 92.1(0.852) | 92.7(0.822) |
| Cov. for $\boldsymbol{\beta}_1^*$ (%) | Batch Means | 92.9(0.812) | 94.1(0.745) | 94.0(0.750) | 92.8(0.817) |
|  | Random Scaling | 94.3(0.733) | 94.6(0.714) | 94.1(0.745) | 93.0(0.806) |
|  | Spectral Variance | 0.061(0.013) | 0.043(0.008) | 0.031(0.005) | 0.022(0.003) |
| Length | Batch Means | 0.061(0.005) | 0.045(0.003) | 0.032(0.002) | 0.023(0.001) |
|  | Random Scaling | 0.085(0.035) | 0.060(0.025) | 0.042(0.016) | 0.030(0.012) |

Table 3: Logistic Regression, $d = 5$

|  |  | $n = 5,000$ | $n = 10,000$ | $n = 20,000$ | $n = 40,000$ |
|---|---|---|---|---|---|
| $\eta = 0.01$ |  |  |  |  |  |
|  | Spectral Variance | 60.0(1.549) | 75.7(1.356) | 84.5(1.144) | 89.9(0.952) |
| Cov. for $\mathbb{E}_{\pi_\eta}\boldsymbol{\beta}_1$ (%) | Batch Means | 32.6(1.482) | 45.4(1.574) | 51.8(1.580) | 57.8(1.561) |
|  | Random Scaling | 86.9(1.066) | 91.0(0.904) | 94.1(0.745) | 95.6(0.648) |
|  | Spectral Variance | 59.4(1.552) | 75.2(1.365) | 84.4(1.147) | 89.7(0.961) |
| Cov. for $\boldsymbol{\beta}_1^*$ (%) | Batch Means | 27.8(1.416) | 40.7(1.553) | 47.9(1.579) | 55.9(1.570) |
|  | Random Scaling | 85.7(1.107) | 89.8(0.957) | 93.0(0.806) | 95.1(0.682) |
|  | Spectral Variance | 0.079(0.022) | 0.068(0.016) | 0.055(0.010) | 0.042(0.006) |
| Length | Batch Means | 0.033(0.007) | 0.028(0.005) | 0.023(0.003) | 0.018(0.002) |
|  | Random Scaling | 0.175(0.080) | 0.129(0.056) | 0.092(0.040) | 0.063(0.026) |
| $\eta = 0.1$ |  |  |  |  |  |
|  | Spectral Variance | 91.2(0.895) | 91.9(0.862) | 91.3(0.891) | 92.3(0.843) |
| Cov. for $\mathbb{E}_{\pi_\eta}\boldsymbol{\beta}_1$ (%) | Batch Means | 76.8(1.334) | 83.7(1.168) | 85.9(1.100) | 87.8(1.034) |
|  | Random Scaling | 95.1(0.682) | 93.2(0.796) | 95.2(0.675) | 94.8(0.702) |
|  | Spectral Variance | 88.2(1.020) | 87.9(1.031) | 85.4(1.116) | 80.1(1.262) |
| Cov. for $\boldsymbol{\beta}_1^*$ (%) | Batch Means | 77.3(1.324) | 80.6(1.250) | 81.8(1.220) | 79.3(1.281) |
|  | Random Scaling | 94.3(0.733) | 92.8(0.817) | 94.1(0.745) | 91.3(0.891) |
|  | Spectral Variance | 0.129(0.028) | 0.094(0.018) | 0.067(0.012) | 0.048(0.007) |
| Length | Batch Means | 0.090(0.009) | 0.071(0.006) | 0.056(0.004) | 0.042(0.002) |
|  | Random Scaling | 0.180(0.079) | 0.129(0.053) | 0.092(0.039) | 0.066(0.027) |

interesting to consider whether such proofs can be simplified by regarding these SA problems as time-inhomogeneous Markov chains and applying techniques in the Markov chain setting.

## Acknowledgments and Disclosure of Funding

This work has been supported by the National Key Research and Development Project of China (No. 2020AAA0104400) and the National Natural Science Foundation of China (No. 12271011).

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
