# A Missing Proofs

## A.1 Proof of Theorem 2.1

Theorem 2.1 is a direct extension of the main theorem in [8].

**Theorem A.1** ([8]). *Let $f \in L_\pi^2$ with $\int f(x)\pi(\mathrm{d}x) = 0$. If there exists $0 < \alpha < \frac{1}{2}$ such that*

$$\left\| \sum_{k=0}^{n-1} P^k f \right\|_{L_\pi^2} = O(n^\alpha), \tag{17}$$

*then $\sigma_f^2 := \lim_n \frac{1}{n} \mathbb{E}_\pi(S_n(f)^2)$ exists and is finite, and for $\pi$-almost every point $x \in S$ the sequence $n^{-\frac{1}{2}} S_n(f)$ converges in distribution, under the probability measure $\mathbb{P}_x$, to the Gaussian distribution $\mathcal{N}(0, \sigma_f^2)$. Furthermore, the process $(Y_n(t))_{0 \le t \le 1}$ converges weakly to $(\sigma_f B(t))_{0 \le t \le 1}$ on the Skorokhod space $D[0, 1]$, where $B = (B(t))_{t \ge 0}$ is the standard Brownian motion.*

Note that Theorem A.1 is for univariate functionals of a Markov chain. We can use the Cramér-Wold device to extend it to a multivariate version. To this end, for a square-integrable multivariate function $\boldsymbol{f} \colon S \to \mathbb{R}^d$, we apply Theorem A.1 with the univariate function $f = \boldsymbol{v}^\top \boldsymbol{f}$ for every $\boldsymbol{v} \in \mathbb{R}^d$. Clearly, we have $f \in L_\pi^2$, $\int f(x)\pi(\mathrm{d}x) = 0$, and $\left\| \sum_{k=0}^{n-1} P^k f \right\| \le \|\boldsymbol{v}\| \left\| \sum_{k=0}^{n-1} P^k \boldsymbol{f} \right\| = O(n^\alpha)$. Applying the Cramér-Wold device concludes the proof.

## A.2 Proof of Proposition 2.1

It suffices to show the proposition holds when $\boldsymbol{f}$ is a univariate function (i.e., $m = 1$), since extending it to a multivariate case only requires multiplying the left-hand side of (2) by the output dimension $m$, which is independent of $n$. Using the Wasserstein contraction (3) with $\mu = \delta_x$ and $\nu = \pi$ together with Jensen's inequality, we have for any $k \ge 0$,

$$\mathcal{W}_{p,1}(\delta_x P^k, \pi) \le \mathcal{W}_{p,q}(\delta_x P^k, \pi) = \mathcal{W}_{p,q}(\delta_x P^k, \pi P^k) \le \gamma^k \mathcal{W}_{p,q}(\delta_x, \pi).$$

The Kantorovich-Rubinstein duality for the Wasserstein distance $\mathcal{W}_{p,1}$ implies that

$$\mathcal{W}_{p,1}(\delta_x P^k, \pi) = \sup_{\mathrm{Lip}_p(g) \le 1} \left| P^k g(x) - \mathbb{E}_\pi g \right| \ge \frac{1}{\mathrm{Lip}_p(f)} \left| P^k f(x) - \mathbb{E}_\pi f \right| = \frac{1}{\mathrm{Lip}_p(f)} \left| P^k f(x) \right|.$$

Therefore, we have

$$\left\| \sum_{k=0}^{n-1} P^k f \right\|_{L_\pi^2}^2 = \mathbb{E}_\pi \left| \sum_{k=0}^{n-1} P^k f(X) \right|^2 \le \mathbb{E}_\pi \left( \sum_{k=0}^{n-1} \left| P^k f(X) \right| \right)^2$$

$$\le \mathrm{Lip}_p(f)^2 \mathbb{E}_\pi \left( \sum_{k=0}^{n-1} \gamma^k \mathcal{W}_{p,q}(\delta_X, \pi) \right)^2$$

$$\le \frac{\mathrm{Lip}_p(f)^2}{(1 - \gamma)^2} \mathbb{E}_\pi \left( \mathcal{W}_{p,q}^2(\delta_X, \pi) \right) = O(1),$$

and thus (2) holds. The last equality follows from the fact that $\mathcal{W}_{p,q}(\delta(\cdot), \pi) \in L_\pi^2$.

## A.3 Proof of Theorem 3.1

The existence and uniqueness of the stationary distribution $\pi_\eta$ and its moments are given by Lemma 11 in [20]. In addition, it states that if $\{\boldsymbol{A}_t\}_{t \ge 1}$ and $\{\boldsymbol{b}_t\}_{t \ge 1}$ have i.i.d. entries, then the covariance matrix $\boldsymbol{\Lambda}_\eta^* := \mathrm{cov}_{\pi_\eta}(\boldsymbol{\theta})$ is the unique solution to the following matrix equation

$$\boldsymbol{A}\boldsymbol{\Lambda} + \boldsymbol{\Lambda}\boldsymbol{A}^\top - \eta \boldsymbol{A}\boldsymbol{\Lambda}\boldsymbol{A}^\top - \eta \mathbb{E}(\boldsymbol{\Xi}\boldsymbol{\Lambda}\boldsymbol{\Xi}^\top) = \eta \boldsymbol{\Sigma}^*, \tag{18}$$

where $\boldsymbol{\Sigma}^* := \mathrm{cov}(\boldsymbol{\Xi}\boldsymbol{\theta}^* + \boldsymbol{\xi})$. Here, $\boldsymbol{\Xi}$ and $\boldsymbol{\xi}$ are independent copies whose distributions are the same as $\{\boldsymbol{\Xi}_t\}_{t \ge 1}$ and $\{\boldsymbol{\xi}_t\}_{t \ge 1}$.

For $k \geq 1$, we have

$$
\begin{aligned}
\mathrm{cov}_{\pi_\eta}(\boldsymbol{\theta}_0, \boldsymbol{\theta}_k) &= \mathbb{E}_{\pi_\eta}[(\boldsymbol{\theta}_0 - \boldsymbol{\theta}^*)(\boldsymbol{\theta}_k - \boldsymbol{\theta}^*)^\top] \\
&= \mathbb{E}_{\pi_\eta}[(\boldsymbol{\theta}_0 - \boldsymbol{\theta}^*)((\boldsymbol{I}_d - \eta \boldsymbol{A}_k)(\boldsymbol{\theta}_{k-1} - \boldsymbol{\theta}^*) - \eta(\boldsymbol{A}_k \boldsymbol{\theta}^* - \boldsymbol{b}_k))^\top] \\
&= \mathbb{E}_{\pi_\eta}\left[\mathbb{E}[(\boldsymbol{\theta}_0 - \boldsymbol{\theta}^*)((\boldsymbol{I}_d - \eta \boldsymbol{A}_k)(\boldsymbol{\theta}_{k-1} - \boldsymbol{\theta}^*) - \eta(\boldsymbol{A}_k \boldsymbol{\theta}^* - \boldsymbol{b}_k))^\top | \mathcal{F}_{k-1}]\right] \\
&= \mathbb{E}_{\pi_\eta}[(\boldsymbol{\theta}_0 - \boldsymbol{\theta}^*)(\boldsymbol{\theta}_{k-1} - \boldsymbol{\theta}^*)^\top (\boldsymbol{I}_d - \eta \boldsymbol{A})^\top] \\
&= \mathrm{cov}_{\pi_\eta}(\boldsymbol{\theta}_0, \boldsymbol{\theta}_{k-1})(\boldsymbol{I}_d - \eta \boldsymbol{A})^\top.
\end{aligned}
$$

Therefore, the covariance matrix $\boldsymbol{\Sigma}_{\pi_\eta}$ is

$$
\begin{aligned}
\boldsymbol{\Sigma}_{\pi_\eta} &= \mathrm{cov}_{\pi_\eta}(\boldsymbol{\theta}_0, \boldsymbol{\theta}_0) + \sum_{k=1}^{\infty}\left(\mathrm{cov}_{\pi_\eta}(\boldsymbol{\theta}_0, \boldsymbol{\theta}_k) + \mathrm{cov}_{\pi_\eta}(\boldsymbol{\theta}_0, \boldsymbol{\theta}_k)^\top\right) \\
&= \boldsymbol{\Lambda}_\eta^* \left(\sum_{k=0}^{\infty}(\boldsymbol{I}_d - \eta \boldsymbol{A})^k\right)^\top + \left(\sum_{k=0}^{\infty}(\boldsymbol{I}_d - \eta \boldsymbol{A})^k\right) \boldsymbol{\Lambda}_\eta^* - \boldsymbol{\Lambda}_\eta^* \\
&= \eta^{-1}\boldsymbol{\Lambda}_\eta^*(\boldsymbol{A}^{-1})^\top + \eta^{-1}\boldsymbol{A}^{-1}\boldsymbol{\Lambda}_\eta^* - \boldsymbol{\Lambda}_\eta^* \\
&= \boldsymbol{A}^{-1}(\boldsymbol{\Sigma}^* + \mathbb{E}(\boldsymbol{\Xi}\boldsymbol{\Lambda}\boldsymbol{\Xi}^\top))(\boldsymbol{A}^{-1})^\top,
\end{aligned}
$$

where the last line follows from (18). The form of $\boldsymbol{\Sigma}_{\pi_\eta}$ matches that in [20].

Finally, since the LSA with a Hurwitz matrix $\boldsymbol{A}$ follows a contraction in the $\mathcal{W}_{2,2}$ distance, the FCLT follows directly from Proposition 2.1 and Theorem 2.1.

## A.4 Proof of Theorem 3.2

We first prove the existence and uniqueness of the stationary distribution. Since $L^2(\mathbb{R}^d)$ is separable and complete, the Wasserstein space $\mathcal{P}_{2,2}$ is complete [27]. Denote $\mathcal{L}(X)$ as the distribution of the random variable $X$. Then it suffices to show $\{\mathcal{L}(\boldsymbol{\theta}_t)\}_{t \geq 0}$ is a Cauchy sequence in this space.

Suppose $\boldsymbol{\theta}_0 \sim \mu$, and take any positive integer $N > 0$. For any $k \geq N$ and $l \geq 0$, we need to bound the Wasserstein distance $\mathcal{W}_{2,2}(\mathcal{L}(\boldsymbol{\theta}_k), \mathcal{L}(\boldsymbol{\theta}_{k+l}))$. Consider the process (9) started from $\boldsymbol{\theta}_0^{(1)} \sim \mathcal{L}(\boldsymbol{\theta}_0)$ and $\boldsymbol{\theta}_0^{(2)} \sim \mathcal{L}(\boldsymbol{\theta}_l)$. In the first $k$ iterations, we couple the two processes with the same stochastic gradient sources $\xi_1, \xi_2, \ldots, \xi_k$. Lemma B.2 and Lemma B.1 imply that

$$
\begin{aligned}
\mathcal{W}_{2,2}^2(\mathcal{L}(\boldsymbol{\theta}_k), \mathcal{L}(\boldsymbol{\theta}_{k+l})) &= \mathcal{W}_{2,2}^2(\mathcal{L}(\boldsymbol{\theta}_k^{(1)}), \mathcal{L}(\boldsymbol{\theta}_k^{(2)})) \\
&\leq (1 - \eta m)^k \mathcal{W}_{2,2}^2(\mathcal{L}(\boldsymbol{\theta}_0^{(1)}), \mathcal{L}(\boldsymbol{\theta}_0^{(2)})) \\
&\leq (1 - \eta m)^k \mathbb{E}\|\boldsymbol{\theta}_l - \boldsymbol{\theta}_0\|^2 \\
&\leq 4(1 - \eta m)^k \sup_{t \geq 0} \mathbb{E}\|\boldsymbol{\theta}_t - \boldsymbol{\theta}^*\|^2 \\
&\leq 4(1 - \eta m)^N \left[\mathbb{E}\|\boldsymbol{\theta}_0 - \boldsymbol{\theta}^*\|^2 + \frac{\eta d}{m}\left(\|\boldsymbol{S}\| + \frac{C}{2}\right)\right].
\end{aligned}
$$

As $N \to \infty$, $\mathcal{W}_{2,2}^2(\mathcal{L}(\boldsymbol{\theta}_k), \mathcal{L}(\boldsymbol{\theta}_{k+l})) \to 0$, and therefore $\{\mathcal{L}(\boldsymbol{\theta}_t)\}_{t \geq 0}$ is a Cauchy sequence in $\mathcal{P}_{2,2}$ and its limit $\pi_\eta$ exists.

The uniqueness follows directly from Lemma B.2. More precisely, let $\boldsymbol{\theta}_0^{(1)} \sim \pi_\eta^{(1)}$ and $\boldsymbol{\theta}_0^{(1)} \sim \pi_\eta^{(2)}$ where $\pi_\eta^{(1)}$ and $\pi_\eta^{(2)}$ are two different stationary distributions. Then $\mathcal{W}_{2,2}^2(\pi_\eta^{(1)}, \pi_\eta^{(2)}) \leq (1 - \eta m)\mathcal{W}_{2,2}^2(\pi_\eta^{(1)}, \pi_\eta^{(2)})$ implies $\mathcal{W}_{2,2}(\pi_\eta^{(1)}, \pi_\eta^{(2)}) = 0$.

Next we bound $\mathbb{E}_{\pi_\eta}\|\boldsymbol{\theta} - \boldsymbol{\theta}^*\|^2$. Since $\boldsymbol{\theta}$ and $\boldsymbol{\theta} - \eta \nabla f(\boldsymbol{\theta}, \xi)$ have the same distribution $\pi_\eta$, we have

$$
\begin{aligned}
\mathbb{E}_{\pi_\eta}\|\boldsymbol{\theta} - \boldsymbol{\theta}^*\|^2 &= \mathbb{E}_{\pi_\eta}\|\boldsymbol{\theta} - \eta \nabla f(\boldsymbol{\theta}, \xi) - \boldsymbol{\theta}^*\|^2 \\
&= \mathbb{E}_{\pi_\eta}\|\boldsymbol{\theta} - \eta \nabla f(\boldsymbol{\theta}) - \boldsymbol{\theta}^* - \eta \varepsilon(\boldsymbol{\theta}, \xi)\|^2 \\
&= \mathbb{E}_{\pi_\eta}\|\boldsymbol{\theta} - \eta \nabla f(\boldsymbol{\theta}) - \boldsymbol{\theta}^*\|^2 + \eta^2 \mathbb{E}_{\pi_\eta}\|\varepsilon(\boldsymbol{\theta}, \xi)\|^2 \\
&\leq (1 - \eta m)^2 \mathbb{E}_{\pi_\eta}\|\boldsymbol{\theta} - \boldsymbol{\theta}^*\|^2 + \eta^2 \mathbb{E}_{\pi_\eta}\|\varepsilon(\boldsymbol{\theta}, \xi)\|^2,
\end{aligned}
$$

and therefore

$$\mathbb{E}_{\pi_\eta}\|\boldsymbol{\theta} - \boldsymbol{\theta}^*\|^2 \leq \frac{\eta}{2m - \eta m^2}\mathbb{E}_{\pi_\eta}\|\varepsilon(\boldsymbol{\theta}, \xi)\|^2 \leq \frac{\eta}{m}\mathbb{E}_{\pi_\eta}\|\varepsilon(\boldsymbol{\theta}, \xi)\|^2.$$

Finally, we establish the FCLT by verifying conditions in Theorem 2.1. Let $\boldsymbol{\mu}_\eta = \mathbb{E}_{\pi_\eta}\boldsymbol{\theta}$ and $\boldsymbol{f}(\boldsymbol{\theta}) = \boldsymbol{\theta} - \boldsymbol{\mu}_\eta \in L^2_{\pi_\eta}$. Obviously, $\mathbb{E}_{\pi_\eta}\boldsymbol{f}(\boldsymbol{\theta}) = 0$. In addition, Lemma B.2 has already shown the $(1 - \eta m)^{1/2}$-contraction of the transition kernel $P$ in the $\mathcal{W}_{2,2}$ distance and thus the condition (2) holds using Proposition 2.1. The FCLT (11) then follows from Theorem 2.1.

### A.5   Proof of Theorem 3.3

The proof of the existence and uniqueness of the stationary distribution $\mathcal{Q}_\eta$ is similar to the proof in Theorem 3.2. More precisely, we prove that $\{\mathcal{L}(\boldsymbol{Q}_t)\}_{t \geq 0}$ is a Cauchy sequence in the complete space $\mathcal{P}_{\infty,1}$.

Suppose $\boldsymbol{Q}_0 \sim \mu$, and take any positive integer $N > 0$. For any $k \geq N$ and $l \geq 0$, we need to bound the Wasserstein distance $\mathcal{W}_{\infty,1}(\mathcal{L}(\boldsymbol{Q}_k), \mathcal{L}(\boldsymbol{Q}_{k+l}))$. Consider the process (12) started from $\boldsymbol{Q}_0^{(1)} \sim \mathcal{L}(\boldsymbol{Q}_0)$ and $\boldsymbol{Q}_0^{(2)} \sim \mathcal{L}(\boldsymbol{Q}_l)$. In the first $k$ iterations, we couple the two processes with the same randomness, i.e., with the same $r_t(s, a)$ and $s_t(s, a)$ for every state-action pair $(s, a)$. Lemma B.3 implies that

$$\begin{aligned}
\mathcal{W}_{\infty,1}(\mathcal{L}(\boldsymbol{Q}_k), \mathcal{L}(\boldsymbol{Q}_{k+l})) &= \mathcal{W}_{\infty,1}(\mathcal{L}(\boldsymbol{Q}_k^{(1)}), \mathcal{L}(\boldsymbol{Q}_k^{(2)})) \\
&\leq (1 - \eta + \eta\gamma)^k \mathcal{W}_{\infty,1}(\mathcal{L}(\boldsymbol{Q}_0^{(1)}), \mathcal{L}(\boldsymbol{Q}_0^{(2)})) \\
&\leq (1 - \eta + \eta\gamma)^N \mathbb{E}\|\boldsymbol{Q}_l - \boldsymbol{Q}_0\|_\infty.
\end{aligned}$$

Since $0 \leq r_t(s, a) \leq 1$, for any $l \geq 0$,

$$\mathbb{E}\|\boldsymbol{Q}_{l+1}\|_\infty = \mathbb{E}\|(1 - \eta)\boldsymbol{Q}_l + \eta\widehat{\mathcal{T}}_{l+1}(\boldsymbol{Q}_l)\|_\infty \leq (1 - \eta + \eta\gamma)\mathbb{E}\|\boldsymbol{Q}_l\|_\infty + \eta.$$

By induction, we have for any $l \geq 0$,

$$\mathbb{E}\|\boldsymbol{Q}_l\|_\infty \leq (1 - \eta + \eta\gamma)^l \mathbb{E}\|\boldsymbol{Q}_0\|_\infty + \frac{\eta}{1 - (1 - \eta + \eta\gamma)} = (1 - \eta + \eta\gamma)^l \mathbb{E}\|\boldsymbol{Q}_0\|_\infty + \frac{1}{1 - \gamma},$$

and thus $\mathbb{E}\|\boldsymbol{Q}_l - \boldsymbol{Q}_0\|_\infty$ is uniformly bounded for all $l \geq 0$. As $N \to \infty$, we have $\mathcal{W}_{\infty,1}(\mathcal{L}(\boldsymbol{Q}_k), \mathcal{L}(\boldsymbol{Q}_{k+l})) \to 0$, and therefore $\{\mathcal{L}(\boldsymbol{Q}_t)\}_{t \geq 0}$ is a Cauchy sequence in $\mathcal{P}_{\infty,1}$ and its limit $\mathcal{Q}_\eta$ exists. As for the uniqueness, suppose $\boldsymbol{Q}_0^{(1)} \sim \mathcal{Q}_\eta^{(1)}$ and $\boldsymbol{Q}_0^{(2)} \sim \mathcal{Q}_\eta^{(2)}$. Then From Lemma B.3, $\mathcal{W}_{\infty,1}(\mathcal{Q}_\eta^{(1)}, \mathcal{Q}_\eta^{(2)}) \leq (1 - \eta + \eta\gamma)\mathcal{W}_{\infty,1}(\mathcal{Q}_\eta^{(1)}, \mathcal{Q}_\eta^{(2)})$ implies $\mathcal{W}_{\infty,1}(\mathcal{Q}_\eta^{(1)}, \mathcal{Q}_\eta^{(2)}) = 0$.

Next we characterize the order of $\mathbb{E}_{\mathcal{Q}_\eta}\|\boldsymbol{Q} - \boldsymbol{Q}^*\|_\infty$. This follows from Theorem 2.1 in [2], where letting $k \to \infty$ gives the claimed result.

Finally, we establish the FCLT for Q-learning iterates by verifying conditions in Theorem 2.1. Let $\boldsymbol{\mu}_\eta = \mathbb{E}_{\mathcal{Q}_\eta}\boldsymbol{Q}$ and $\boldsymbol{f}(\boldsymbol{Q}) = \boldsymbol{Q} - \boldsymbol{\mu}_\eta \in L^2_{\mathcal{Q}_\eta}$. Obviously, $\mathbb{E}_{\mathcal{Q}_\eta}\boldsymbol{f}(\boldsymbol{Q}) = 0$. In addition, Lemma B.3 has already shown the $(1 - \eta + \eta\gamma)$-contraction of the transition kernel $P$ in the $\mathcal{W}_{\infty,1}$ distance and thus the condition (2) holds using Proposition 2.1. The FCLT (14) then follows from Theorem 2.1.

## B   Auxiliary Lemmas

**Lemma B.1.** *Suppose Assumptions 3.3 and 3.4 hold for the SGD iterates* (9). *For any $\eta < \min\left\{\frac{1}{L}, \frac{2m}{2m^2 + 3Cd}\right\}$ and any $t \geq 0$, we have*

$$\mathbb{E}\|\boldsymbol{\theta}_t - \boldsymbol{\theta}^*\|^2 \leq (1 - \eta m)^t \mathbb{E}\|\boldsymbol{\theta}_0 - \boldsymbol{\theta}^*\|^2 + \frac{\eta d}{m}\left(\|\boldsymbol{S}\| + \frac{C}{2}\right). \tag{19}$$

*Proof.* We have

$$\mathbb{E}\|\boldsymbol{\theta}_{t+1} - \boldsymbol{\theta}^*\|^2 = \mathbb{E}\|\boldsymbol{\theta}_t - \eta\nabla f(\boldsymbol{\theta}_t, \xi_{t+1}) - \boldsymbol{\theta}^*\|^2$$
$$= \mathbb{E}\|\boldsymbol{\theta}_t - \eta\nabla f(\boldsymbol{\theta}_t) - \boldsymbol{\theta}^* - \eta\varepsilon(\boldsymbol{\theta}_t, \xi_{t+1})\|^2$$
$$= \mathbb{E}\|\boldsymbol{\theta}_t - \eta\nabla f(\boldsymbol{\theta}_t) - \boldsymbol{\theta}^*\|^2 + \eta^2\mathbb{E}\|\varepsilon(\boldsymbol{\theta}_t, \xi_{t+1})\|^2$$
$$\qquad - 2\eta\mathbb{E}\langle\boldsymbol{\theta}_t - \eta\nabla f(\boldsymbol{\theta}_t) - \boldsymbol{\theta}^*, \varepsilon(\boldsymbol{\theta}_t, \xi_{t+1})\rangle$$
$$=: T_1 + T_2 + T_3.$$

For the first term $T_1$,

$$\mathbb{E}\|\boldsymbol{\theta}_t - \eta\nabla f(\boldsymbol{\theta}_t) - \boldsymbol{\theta}^*\|^2 = \mathbb{E}\|(\boldsymbol{\theta}_t - \boldsymbol{\theta}^*) - \eta(\nabla f(\boldsymbol{\theta}_t) - \nabla f(\boldsymbol{\theta}^*))\|^2$$
$$= \mathbb{E}\left\|\int_0^1 \left[\boldsymbol{I}_d - \eta\nabla^2 f(s\boldsymbol{\theta}_t + (1-s)\boldsymbol{\theta}^*)\right](\boldsymbol{\theta}_t - \boldsymbol{\theta}^*)\mathrm{d}s\right\|^2$$
$$\leq \mathbb{E}\left(\int_0^1 \left\|\left[\boldsymbol{I}_d - \eta\nabla^2 f(s\boldsymbol{\theta}_t + (1-s)\boldsymbol{\theta}^*)\right](\boldsymbol{\theta}_t - \boldsymbol{\theta}^*)\right\|\mathrm{d}s\right)^2$$
$$\leq \mathbb{E}\left(\int_0^1 \left\|\boldsymbol{I}_d - \eta\nabla^2 f(s\boldsymbol{\theta}_t + (1-s)\boldsymbol{\theta}^*)\right\|\|\boldsymbol{\theta}_t - \boldsymbol{\theta}^*\|\mathrm{d}s\right)^2$$
$$\leq (1 - \eta m)^2\mathbb{E}\|\boldsymbol{\theta}_t - \boldsymbol{\theta}^*\|^2,$$

where the last inequality follows from $\eta \leq \frac{1}{L}$ and $m\boldsymbol{I}_d \preccurlyeq \nabla^2 f(z) \preccurlyeq L\boldsymbol{I}_d$ for any $z \in \mathbb{R}^d$. For the second term $T_2$,

$$\eta^2\mathbb{E}\|\varepsilon(\boldsymbol{\theta}_t, \xi_{t+1})\|^2 = \eta^2\mathbb{E}\left[\mathbb{E}[\varepsilon(\boldsymbol{\theta}_t, \xi_{t+1})^\top\varepsilon(\boldsymbol{\theta}_t, \xi_{t+1})|\mathcal{F}_t]\right]$$
$$= \eta^2\mathbb{E}\left[\mathrm{Tr}\left(\mathbb{E}[\varepsilon(\boldsymbol{\theta}_t, \xi_{t+1})\varepsilon(\boldsymbol{\theta}_t, \xi_{t+1})^\top|\mathcal{F}_t]\right)\right]$$
$$\leq \eta^2 d\mathbb{E}\left[\left\|\mathbb{E}[\varepsilon(\boldsymbol{\theta}_t, \xi_{t+1})\varepsilon(\boldsymbol{\theta}_t, \xi_{t+1})^\top|\mathcal{F}_t]\right\|\right]$$
$$\leq \eta^2 d\mathbb{E}\left[\|\boldsymbol{S}\| + C\|\boldsymbol{\theta}_t - \boldsymbol{\theta}^*\| + C\|\boldsymbol{\theta}_t - \boldsymbol{\theta}^*\|^2\right]$$
$$\leq \eta^2 d\left(\|\boldsymbol{S}\| + \frac{C}{2}\right) + \frac{3C\eta^2 d}{2}\mathbb{E}\|\boldsymbol{\theta}_t - \boldsymbol{\theta}^*\|^2.$$

The third term $T_3$ equals

$$-2\eta\mathbb{E}\langle\boldsymbol{\theta}_t - \eta\nabla f(\boldsymbol{\theta}_t) - \boldsymbol{\theta}^*, \varepsilon(\boldsymbol{\theta}_t, \xi_{t+1})\rangle = -2\eta\mathbb{E}\left[\mathbb{E}[\langle\boldsymbol{\theta}_t - \eta\nabla f(\boldsymbol{\theta}_t) - \boldsymbol{\theta}^*, \varepsilon(\boldsymbol{\theta}_t, \xi_{t+1})\rangle|\mathcal{F}_t]\right] = 0.$$

Putting these together yields

$$\mathbb{E}\|\boldsymbol{\theta}_{t+1} - \boldsymbol{\theta}^*\|^2 \leq \left((1 - \eta m)^2 + \frac{3C\eta^2 d}{2}\right)\mathbb{E}\|\boldsymbol{\theta}_t - \boldsymbol{\theta}^*\|^2 + \eta^2 d\left(\|\boldsymbol{S}\| + \frac{C}{2}\right)$$
$$\leq (1 - \eta m)\mathbb{E}\|\boldsymbol{\theta}_t - \boldsymbol{\theta}^*\|^2 + \eta^2 d\left(\|\boldsymbol{S}\| + \frac{C}{2}\right),$$

since $\eta < \frac{2m}{2m^2 + 3Cd}$. By induction, we obtain for any $t \geq 0$,

$$\mathbb{E}\|\boldsymbol{\theta}_t - \boldsymbol{\theta}^*\|^2 \leq (1 - \eta m)^t\mathbb{E}\|\boldsymbol{\theta}_0 - \boldsymbol{\theta}^*\|^2 + \frac{\eta d}{m}\left(\|\boldsymbol{S}\| + \frac{C}{2}\right).$$

$\square$

**Lemma B.2.** *Suppose Assumptions 3.3 and 3.4 hold for the SGD iterates $(\boldsymbol{\theta}_t^{(1)})_{t\geq 0}$ and $(\boldsymbol{\theta}_t^{(2)})_{t\geq 0}$, following (9) with the same stochastic gradient sources $\xi_1, \xi_2, \ldots$. For any $\eta < \min\left\{\frac{1}{L}, \frac{m}{m^2 + 4L^2}\right\}$ and any $t \geq 0$, we have*

$$\mathbb{E}\|\boldsymbol{\theta}_{t+1}^{(1)} - \boldsymbol{\theta}_{t+1}^{(2)}\|^2 \leq (1 - \eta m)\mathbb{E}\|\boldsymbol{\theta}_t^{(1)} - \boldsymbol{\theta}_t^{(2)}\|^2. \tag{20}$$

*In other words, the process (9) admits a contraction in the $\mathcal{W}_{2,2}$ distance,*

$$\mathcal{W}_{2,2}^2(\mathcal{L}(\boldsymbol{\theta}_{t+1}^{(1)}), \mathcal{L}(\boldsymbol{\theta}_{t+1}^{(2)})) \leq (1 - \eta m)\mathcal{W}_{2,2}^2(\mathcal{L}(\boldsymbol{\theta}_t^{(1)}), \mathcal{L}(\boldsymbol{\theta}_t^{(2)})). \tag{21}$$

*Proof.* Since $(\boldsymbol{\theta}_t^{(1)})_{t\geq 0}$ and $(\boldsymbol{\theta}_t^{(2)})_{t\geq 0}$ are coupled through the same stochastic gradient sources $\xi_1, \xi_2, \ldots$, we have

$$
\begin{aligned}
\mathbb{E}\|\boldsymbol{\theta}_{t+1}^{(1)} - \boldsymbol{\theta}_{t+1}^{(2)}\|^2 &= \mathbb{E}\|\boldsymbol{\theta}_t^{(1)} - \eta\nabla f(\boldsymbol{\theta}_t^{(1)}, \xi_{t+1}) - \boldsymbol{\theta}_t^{(2)} + \eta\nabla f(\boldsymbol{\theta}_t^{(2)}, \xi_{t+1})\|^2 \\
&= \mathbb{E}\|\boldsymbol{\theta}_t^{(1)} - \eta\nabla f(\boldsymbol{\theta}_t^{(1)}) - \boldsymbol{\theta}_t^{(2)} + \eta\nabla f(\boldsymbol{\theta}_t^{(2)}) - \eta\varepsilon(\boldsymbol{\theta}_t^{(1)}, \xi_{t+1}) + \eta\varepsilon(\boldsymbol{\theta}_t^{(2)}, \xi_{t+1})\|^2 \\
&= \mathbb{E}\|\boldsymbol{\theta}_t^{(1)} - \eta\nabla f(\boldsymbol{\theta}_t^{(1)}) - \boldsymbol{\theta}_t^{(2)} + \eta\nabla f(\boldsymbol{\theta}_t^{(2)})\|^2 + \eta^2\mathbb{E}\|\varepsilon(\boldsymbol{\theta}_t^{(1)}, \xi_{t+1}) - \varepsilon(\boldsymbol{\theta}_t^{(2)}, \xi_{t+1})\|^2 \\
&\quad - 2\eta\mathbb{E}\langle\boldsymbol{\theta}_t^{(1)} - \eta\nabla f(\boldsymbol{\theta}_t^{(1)}) - \boldsymbol{\theta}_t^{(2)} + \eta\nabla f(\boldsymbol{\theta}_t^{(2)}), \varepsilon(\boldsymbol{\theta}_t^{(1)}, \xi_{t+1}) - \varepsilon(\boldsymbol{\theta}_t^{(2)}, \xi_{t+1})\rangle \\
&=: T_1 + T_2 + T_3.
\end{aligned}
$$

For the first term $T_1$,

$$
\begin{aligned}
&\mathbb{E}\|\boldsymbol{\theta}_t^{(1)} - \eta\nabla f(\boldsymbol{\theta}_t^{(1)}) - \boldsymbol{\theta}_t^{(2)} + \eta\nabla f(\boldsymbol{\theta}_t^{(2)})\|^2 \\
&= \mathbb{E}\left\|\int_0^1 \left[\boldsymbol{I}_d - \eta\nabla^2 f(s\boldsymbol{\theta}_t^{(1)} + (1-s)\boldsymbol{\theta}_t^{(2)})\right](\boldsymbol{\theta}_t^{(1)} - \boldsymbol{\theta}_t^{(2)})\mathrm{d}s\right\|^2 \\
&\leq \mathbb{E}\left(\int_0^1 \left\|\left[\boldsymbol{I}_d - \eta\nabla^2 f(s\boldsymbol{\theta}_t^{(1)} + (1-s)\boldsymbol{\theta}_t^{(2)})\right](\boldsymbol{\theta}_t^{(1)} - \boldsymbol{\theta}_t^{(2)})\right\|\mathrm{d}s\right)^2 \\
&\leq \mathbb{E}\left(\int_0^1 \left\|\boldsymbol{I}_d - \eta\nabla^2 f(s\boldsymbol{\theta}_t^{(1)} + (1-s)\boldsymbol{\theta}_t^{(2)})\right\|\left\|\boldsymbol{\theta}_t^{(1)} - \boldsymbol{\theta}_t^{(2)}\right\|\mathrm{d}s\right)^2 \\
&\leq (1 - \eta m)^2\mathbb{E}\|\boldsymbol{\theta}_t^{(1)} - \boldsymbol{\theta}_t^{(2)}\|^2,
\end{aligned}
$$

where the last inequality follows from $\eta \leq \frac{1}{L}$ and $m\boldsymbol{I}_d \preccurlyeq \nabla^2 f(z) \preccurlyeq L\boldsymbol{I}_d$ for any $z \in \mathbb{R}^d$. For the second term $T_2$,

$$
\begin{aligned}
&\eta^2\mathbb{E}\|\varepsilon(\boldsymbol{\theta}_t^{(1)}, \xi_{t+1}) - \varepsilon(\boldsymbol{\theta}_t^{(2)}, \xi_{t+1})\|^2 \\
&= \eta^2\mathbb{E}\|\nabla f(\boldsymbol{\theta}_t^{(1)}, \xi_{t+1}) - \nabla f(\boldsymbol{\theta}_t^{(1)}) - \nabla f(\boldsymbol{\theta}_t^{(2)}, \xi_{t+1}) + \nabla f(\boldsymbol{\theta}_t^{(2)})\|^2 \\
&\leq 2\eta^2\left(\mathbb{E}\|\nabla f(\boldsymbol{\theta}_t^{(1)}, \xi_{t+1}) - \nabla f(\boldsymbol{\theta}_t^{(2)}, \xi_{t+1})\|^2 + \mathbb{E}\|\nabla f(\boldsymbol{\theta}_t^{(1)}) - \nabla f(\boldsymbol{\theta}_t^{(2)})\|^2\right) \\
&\leq 4\eta^2 L^2\mathbb{E}\|\boldsymbol{\theta}_t^{(1)} - \boldsymbol{\theta}_t^{(2)}\|^2,
\end{aligned}
$$

where the last inequality follows from the $L$-smoothness of $f(\cdot, \xi)$ and $f(\cdot)$. The third term $T_3$ equals

$$
\begin{aligned}
&-2\eta\mathbb{E}\langle\boldsymbol{\theta}_t^{(1)} - \eta\nabla f(\boldsymbol{\theta}_t^{(1)}) - \boldsymbol{\theta}_t^{(2)} + \eta\nabla f(\boldsymbol{\theta}_t^{(2)}), \varepsilon(\boldsymbol{\theta}_t^{(1)}, \xi_{t+1}) - \varepsilon(\boldsymbol{\theta}_t^{(2)}, \xi_{t+1})\rangle \\
&= -2\eta\mathbb{E}\left[\mathbb{E}\left[\langle\boldsymbol{\theta}_t^{(1)} - \eta\nabla f(\boldsymbol{\theta}_t^{(1)}) - \boldsymbol{\theta}_t^{(2)} + \eta\nabla f(\boldsymbol{\theta}_t^{(2)}), \varepsilon(\boldsymbol{\theta}_t^{(1)}, \xi_{t+1}) - \varepsilon(\boldsymbol{\theta}_t^{(2)}, \xi_{t+1})\rangle|\mathcal{F}_t\right]\right] = 0.
\end{aligned}
$$

Putting these together yields a contraction

$$
\mathbb{E}\|\boldsymbol{\theta}_{t+1}^{(1)} - \boldsymbol{\theta}_{t+1}^{(2)}\|^2 \leq ((1 - \eta m)^2 + 4\eta^2 L^2)\mathbb{E}\|\boldsymbol{\theta}_t^{(1)} - \boldsymbol{\theta}_t^{(2)}\|^2 \leq (1 - \eta m)\mathbb{E}\|\boldsymbol{\theta}_t^{(1)} - \boldsymbol{\theta}_t^{(2)}\|^2,
$$

whenever $\eta < \frac{m}{m^2 + 4L^2}$. $\qquad\square$

**Lemma B.3.** *Suppose the Q-learning iterates* $(\boldsymbol{Q}_t^{(1)})_{t\geq 0}$ *and* $(\boldsymbol{Q}_t^{(2)})_{t\geq 0}$ *follow the update rule* (12) *and share all randomness in* $r_t$ *and* $s_t$. *For any* $\eta < \frac{1}{1-\gamma}$ *and any* $t \geq 0$, *we have*

$$
\mathbb{E}\|\boldsymbol{Q}_{t+1}^{(1)} - \boldsymbol{Q}_{t+1}^{(2)}\|_\infty \leq (1 - \eta + \eta\gamma)\mathbb{E}\|\boldsymbol{Q}_t^{(1)} - \boldsymbol{Q}_t^{(2)}\|_\infty. \tag{22}
$$

*In other words, the process* (14) *admits a contraction in the* $\mathcal{W}_{\infty,1}$ *distance,*

$$
\mathcal{W}_{\infty,1}(\mathcal{L}(\boldsymbol{Q}_{t+1}^{(1)}), \mathcal{L}(\boldsymbol{Q}_{t+1}^{(2)})) \leq (1 - \eta + \eta\gamma)\mathcal{W}_{\infty,1}(\mathcal{L}(\boldsymbol{Q}_t^{(1)}), \mathcal{L}(\boldsymbol{Q}_t^{(2)})). \tag{23}
$$

*Proof.* We have

$$\mathbb{E}\|\boldsymbol{Q}_{t+1}^{(1)} - \boldsymbol{Q}_{t+1}^{(2)}\|_\infty = \mathbb{E}\|(1-\eta)\boldsymbol{Q}_t^{(1)} + \eta\widehat{\mathcal{T}}_{t+1}(\boldsymbol{Q}_t^{(1)}) - (1-\eta)\boldsymbol{Q}_t^{(2)} - \eta\widehat{\mathcal{T}}_{t+1}(\boldsymbol{Q}_t^{(2)})\|_\infty$$

$$= \mathbb{E}\max_{s\in\mathcal{S},a\in\mathcal{A}}\left|(1-\eta)(\boldsymbol{Q}_t^{(1)}(s,a) - \boldsymbol{Q}_t^{(2)}(s,a))\right.$$

$$\left. + \eta\gamma\left(\max_{a'\in\mathcal{A}}\boldsymbol{Q}_t^{(1)}(s_t,a') - \max_{a'\in\mathcal{A}}\boldsymbol{Q}_t^{(2)}(s_t,a')\right)\right|$$

$$\leq (1-\eta)\mathbb{E}\|\boldsymbol{Q}_t^{(1)} - \boldsymbol{Q}_t^{(2)}\|_\infty$$

$$+ \eta\gamma\mathbb{E}\max_{s\in\mathcal{S},a\in\mathcal{A}}\left|\max_{a'\in\mathcal{A}}\left(\boldsymbol{Q}_t^{(1)}(s_t,a') - \boldsymbol{Q}_t^{(2)}(s_t,a')\right)\right|$$

$$\leq (1-\eta+\eta\gamma)\mathbb{E}\|\boldsymbol{Q}_t^{(1)} - \boldsymbol{Q}_t^{(2)}\|_\infty,$$

where the last inequality follows from the definition of the $\|\cdot\|_\infty$ norm. $\square$

## C  Full experiment results for SGD

In this section, we present full experiment results which are not included in Section 4. Table 4-5 summarize the coverage rates, the CI lengths and their standard errors of three inference methods for linear regression with $d = 5, 20$, respectively. Table 6-7 summarize those for Logistic regression.

Table 4: Linear Regression, $d = 5$

|  |  | $n = 5,000$ | $n = 10,000$ | $n = 20,000$ | $n = 40,000$ |
|---|---|---|---|---|---|
| **$\eta = 0.01$** |  |  |  |  |  |
|  | Spectral Variance | 88.6(1.005) | 91.1(0.900) | 91.5(0.881) | 92.4(0.837) |
| Cov. for $\boldsymbol{\beta}_1^*$ (%) | Batch Means | 69.8(1.451) | 77.8(1.314) | 82.3(1.206) | 86.7(1.073) |
|  | Random Scaling | 92.8(0.817) | 94.6(0.714) | 95.0(0.689) | 94.9(0.695) |
|  | Spectral Variance | 0.050(0.011) | 0.036(0.007) | 0.026(0.004) | 0.018(0.003) |
| Length | Batch Means | 0.030(0.003) | 0.024(0.002) | 0.019(0.001) | 0.015(0.001) |
|  | Random Scaling | 0.074(0.033) | 0.053(0.022) | 0.036(0.014) | 0.026(0.010) |
| **$\eta = 0.05$** |  |  |  |  |  |
|  | Spectral Variance | 91.9(0.862) | 91.6(0.877) | 90.8(0.913) | 91.6(0.877) |
| Cov. for $\boldsymbol{\beta}_1^*$ (%) | Batch Means | 90.0(0.948) | 93.6(0.773) | 92.7(0.822) | 92.0(0.857) |
|  | Random Scaling | 94.9(0.695) | 95.2(0.675) | 95.4(0.662) | 94.1(0.745) |
|  | Spectral Variance | 0.055(0.012) | 0.039(0.007) | 0.028(0.005) | 0.020(0.003) |
| Length | Batch Means | 0.050(0.004) | 0.038(0.002) | 0.027(0.001) | 0.020(0.001) |
|  | Random Scaling | 0.076(0.031) | 0.055(0.023) | 0.039(0.015) | 0.027(0.010) |
| **$\eta = 0.1$** |  |  |  |  |  |
|  | Spectral Variance | 89.5(0.969) | 89.8(0.957) | 92.1(0.852) | 92.7(0.822) |
| Cov. for $\boldsymbol{\beta}_1^*$ (%) | Batch Means | 92.9(0.812) | 94.1(0.745) | 94.0(0.750) | 92.8(0.817) |
|  | Random Scaling | 94.3(0.733) | 94.6(0.714) | 94.1(0.745) | 93.0(0.806) |
|  | Spectral Variance | 0.061(0.013) | 0.043(0.008) | 0.031(0.005) | 0.022(0.003) |
| Length | Batch Means | 0.061(0.005) | 0.045(0.003) | 0.032(0.002) | 0.023(0.001) |
|  | Random Scaling | 0.085(0.035) | 0.060(0.025) | 0.042(0.016) | 0.030(0.012) |

Table 5: Linear Regression, $d = 20$

|  |  | $n = 20,000$ | $n = 50,000$ | $n = 80,000$ | $n = 100,000$ |
|---|---|---|---|---|---|
| **$\eta = 0.01$** |  |  |  |  |  |
|  | Spectral Variance | 91.9(0.862) | 92.5(0.832) | 91.7(0.872) | 93.5(0.779) |
| Cov. for $\beta_1^*$ (%) | Batch Means | 80.3(1.257) | 86.1(1.093) | 88.6(1.005) | 88.9(0.993) |
|  | Random Scaling | 94.1(0.745) | 94.4(0.727) | 95.8(0.634) | 97.0(0.539) |
|  | Spectral Variance | 0.027(0.005) | 0.017(0.002) | 0.013(0.001) | 0.012(0.001) |
| Length | Batch Means | 0.019(0.002) | 0.014(0.001) | 0.012(0.001) | 0.010(0.001) |
|  | Random Scaling | 0.038(0.016) | 0.024(0.010) | 0.019(0.008) | 0.017(0.007) |
| **$\eta = 0.05$** |  |  |  |  |  |
|  | Spectral Variance | 91.7(0.872) | 93.1(0.801) | 93.2(0.796) | 93.3(0.790) |
| Cov. for $\beta_1^*$ (%) | Batch Means | 93.8(0.762) | 94.8(0.007) | 95.1(0.682) | 94.1(0.745) |
|  | Random Scaling | 94.7(0.708) | 95.5(0.655) | 94.8(0.702) | 94.8(0.702) |
|  | Spectral Variance | 0.038(0.007) | 0.024(0.004) | 0.019(0.003) | 0.017(0.002) |
| Length | Batch Means | 0.037(0.002) | 0.024(0.001) | 0.019(0.001) | 0.017(0.001) |
|  | Random Scaling | 0.054(0.022) | 0.033(0.013) | 0.026(0.010) | 0.023(0.009) |

\* SGD does not converge when $\eta = 0.1$, so corresponding results are omitted.

Table 6: Logistic Regression, $d = 5$

|  |  | $n = 5,000$ | $n = 10,000$ | $n = 20,000$ | $n = 40,000$ |
|---|---|---|---|---|---|
| **$\eta = 0.01$** |  |  |  |  |  |
|  | Spectral Variance | 60.0(1.549) | 75.7(1.356) | 84.5(1.144) | 89.9(0.952) |
| Cov. for $\mathbb{E}_{\pi_\eta}\beta_1$ (%) | Batch Means | 32.6(1.482) | 45.4(1.574) | 51.8(1.580) | 57.8(1.561) |
|  | Random Scaling | 86.9(1.066) | 91.0(0.904) | 94.1(0.745) | 95.6(0.648) |
|  | Spectral Variance | 59.4(1.552) | 75.2(1.365) | 84.4(1.147) | 89.7(0.961) |
| Cov. for $\beta_1^*$ (%) | Batch Means | 27.8(1.416) | 40.7(1.553) | 47.9(1.579) | 55.9(1.570) |
|  | Random Scaling | 85.7(1.107) | 89.8(0.957) | 93.0(0.806) | 95.1(0.682) |
|  | Spectral Variance | 0.079(0.022) | 0.068(0.016) | 0.055(0.010) | 0.042(0.006) |
| Length | Batch Means | 0.033(0.007) | 0.028(0.005) | 0.023(0.003) | 0.018(0.002) |
|  | Random Scaling | 0.175(0.080) | 0.129(0.056) | 0.092(0.040) | 0.063(0.026) |
| **$\eta = 0.05$** |  |  |  |  |  |
|  | Spectral Variance | 90.1(0.944) | 91.9(0.862) | 91.3(0.891) | 92.3(0.843) |
| Cov. for $\mathbb{E}_{\pi_\eta}\beta_1$ (%) | Batch Means | 67.3(1.483) | 73.8(1.390) | 78.9(1.290) | 82.7(1.196) |
|  | Random Scaling | 93.9(0.756) | 93.1(0.801) | 93.6(0.773) | 94.1(0.745) |
|  | Spectral Variance | 90.2(0.940) | 91.7(0.872) | 91.1(0.900) | 91.7(0.872) |
| Cov. for $\beta_1^*$ (%) | Batch Means | 67.7(1.478) | 74.7(1.374) | 78.5(1.299) | 81.3(1.233) |
|  | Random Scaling | 94.3(0.733) | 92.7(0.822) | 92.9(0.812) | 90.4(0.931) |
|  | Spectral Variance | 0.116(0.026) | 0.087(0.017) | 0.064(0.011) | 0.046(0.007) |
| Length | Batch Means | 0.066(0.009) | 0.055(0.005) | 0.044(0.003) | 0.035(0.002) |
|  | Random Scaling | 0.167(0.074) | 0.122(0.049) | 0.088(0.037) | 0.064(0.027) |
| **$\eta = 0.1$** |  |  |  |  |  |
|  | Spectral Variance | 91.2(0.895) | 91.9(0.862) | 91.3(0.891) | 92.3(0.843) |
| Cov. for $\mathbb{E}_{\pi_\eta}\beta_1$ (%) | Batch Means | 76.8(1.334) | 83.7(1.168) | 85.9(1.100) | 87.8(1.034) |
|  | Random Scaling | 95.1(0.682) | 93.2(0.796) | 95.2(0.675) | 94.8(0.702) |
|  | Spectral Variance | 88.2(1.020) | 87.9(1.031) | 85.4(1.116) | 80.1(1.262) |
| Cov. for $\beta_1^*$ (%) | Batch Means | 77.3(1.324) | 80.6(1.250) | 81.8(1.220) | 79.3(1.281) |
|  | Random Scaling | 94.3(0.733) | 92.8(0.817) | 94.1(0.745) | 91.3(0.891) |
|  | Spectral Variance | 0.129(0.028) | 0.094(0.018) | 0.067(0.012) | 0.048(0.007) |
| Length | Batch Means | 0.090(0.009) | 0.071(0.006) | 0.056(0.004) | 0.042(0.002) |
|  | Random Scaling | 0.180(0.079) | 0.129(0.053) | 0.092(0.039) | 0.066(0.027) |

Table 7: Logistic Regression, $d = 20$

|  |  | $n = 20,000$ | $n = 50,000$ | $n = 80,000$ | $n = 100,000$ |
|---|---|---|---|---|---|
| **$\eta = 0.01$** | | | | | |
| | Spectral Variance | 80.3(1.257) | 88.4(1.012) | 90.1(0.944) | 91.5(0.881) |
| Cov. for $\mathbb{E}_{\pi_\eta}\boldsymbol{\beta}_1$ (%) | Batch Means | 45.2(1.573) | 49.1(1.580) | 53.5(1.577) | 53.5(1.577) |
| | Random Scaling | 91.5(0.881) | 93.6(0.773) | 93.8(0.762) | 93.7(0.768) |
| | Spectral Variance | 81.2(1.235) | 88.4(1.012) | 90.1(0.944) | 92.5(0.832) |
| Cov. for $\boldsymbol{\beta}_1^*$ (%) | Batch Means | 44.0(1.569) | 51.1(1.580) | 56.9(1.566) | 56.6(1.567) |
| | Random Scaling | 89.4(0.973) | 89.6(0.965) | 88.7(1.001) | 88.2(1.020) |
| | Spectral Variance | 0.054(0.012) | 0.042(0.007) | 0.036(0.006) | 0.032(0.005) |
| Length | Batch Means | 0.021(0.003) | 0.017(0.002) | 0.015(0.001) | 0.015(0.001) |
| | Random Scaling | 0.088(0.040) | 0.061(0.026) | 0.049(0.021) | 0.044(0.018) |
| **$\eta = 0.05$** | | | | | |
| | Spectral Variance | 90.4(0.931) | 92.4(0.837) | 91.2(0.895) | 91.8(0.867) |
| Cov. for $\mathbb{E}_{\pi_\eta}\boldsymbol{\beta}_1$ (%) | Batch Means | 72.1(1.418) | 81.8(1.220) | 84.3(1.150) | 85.1(1.126) |
| | Random Scaling | 93.6(0.773) | 93.9(0.756) | 95.4(0.662) | 95.4(0.662) |
| | Spectral Variance | 91.0(0.904) | 91.8(0.867) | 91.4(0.886) | 89.7(0.961) |
| Cov. for $\boldsymbol{\beta}_1^*$ (%) | Batch Means | 70.7(1.439) | 79.3(1.281) | 80.6(1.250) | 80.3(1.257) |
| | Random Scaling | 92.6(0.827) | 93.6(0.773) | 93.4(0.785) | 93.0(0.806) |
| | Spectral Variance | 0.082(0.016) | 0.053(0.009) | 0.043(0.006) | 0.038(0.005) |
| Length | Batch Means | 0.049(0.004) | 0.037(0.002) | 0.031(0.002) | 0.029(0.001) |
| | Random Scaling | 0.113(0.048) | 0.073(0.030) | 0.058(0.023) | 0.053(0.021) |
| **$\eta = 0.1$** | | | | | |
| | Spectral Variance | 91.6(0.877) | 92.7(0.822) | 92.7(0.822) | 93.2(0.796) |
| Cov. for $\mathbb{E}_{\pi_\eta}\boldsymbol{\beta}_1$ (%) | Batch Means | 81.2(1.235) | 87.2(1.056) | 89.0(0.989) | 89.9(0.952) |
| | Random Scaling | 95.4(0.662) | 93.1(0.801) | 94.3(0.733) | 94.1(0.745) |
| | Spectral Variance | 90.0(0.948) | 85.8(1.103) | 84.1(1.156) | 81.4(1.230) |
| Cov. for $\boldsymbol{\beta}_1^*$ (%) | Batch Means | 78.9(1.290) | 80.5(1.252) | 77.8(1.314) | 76.5(1.340) |
| | Random Scaling | 95.0(0.689) | 93.4(0.785) | 94.7(0.708) | 94.7(0.708) |
| | Spectral Variance | 0.097(0.017) | 0.063(0.010) | 0.050(0.007) | 0.045(0.006) |
| Length | Batch Means | 0.071(0.005) | 0.051(0.003) | 0.043(0.002) | 0.039(0.002) |
| | Random Scaling | 0.133(0.055) | 0.087(0.035) | 0.068(0.027) | 0.062(0.024) |

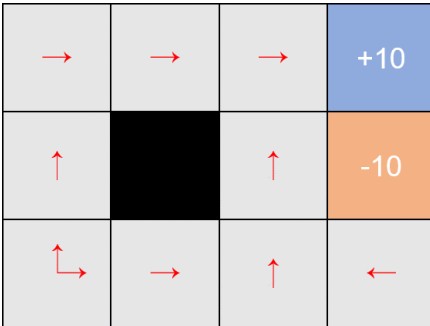

Figure 1: Grid World

# D  Experiment results for Q-Learning

In this section, we consider synchronous Q-learning in a Grid World environment. The environment consists of $3 \times 4$ grids as its state space (see Figure 1). At each grid, the agent can choose to go "up", "down", "left" or "right" into the next grid. If it touches the edge or the block (i.e., the black grid), then it stays still. It stops after arriving at the blue/orange grid. Each immediate reward is $-1$ at grey grids, and $+10/-10$ at the blue/orange grid, respectively. The discount factor is set to $\gamma = 0.9$. Under this deterministic reward setting, the optimal policy at each grid is represented by a red arrow in Figure 1.

Now, we consider randomizing the immediate reward at each grid with a $\mathcal{N}(0, \sigma^2)$ Gaussian noise. We set $\sigma = 2, 4$ and the learning rate $\eta = 0.1, 0.2, 0.3$. The performance of the spectral variance method, batch-means method, and random scaling method are compared in three aspects: the coverage rates for $\mathbb{E}_{\mathcal{Q}_\eta} Q$ and $Q^*$, and the lengths of confidence intervals. Here, the expected Q-value $\mathbb{E}_{\mathcal{Q}_\eta} Q$ is approximated by 500,000 Monte Carlo simulations, and the true Q-value $Q^*$ is derived by Q-learning in the deterministic reward setting. All hyper-parameters for the first two inference methods are the same as in the SGD experiment. The nominal coverage rate is chosen as $95\%$.

Figure 2-3 show the result. First, similar to the SGD experiment, three methods are all valid for inference on $\mathbb{E}_{\mathcal{Q}_\eta} Q$, while invalid for inference on $Q^*$ due to problem-dependent bias. In addition, the random scaling method tends to give wider CI lengths and thus higher coverage rates compared to the nominal rate $95\%$, while the other two often give lower coverage rates. Also, random scaling is usually the first to approach the nominal coverage rate.

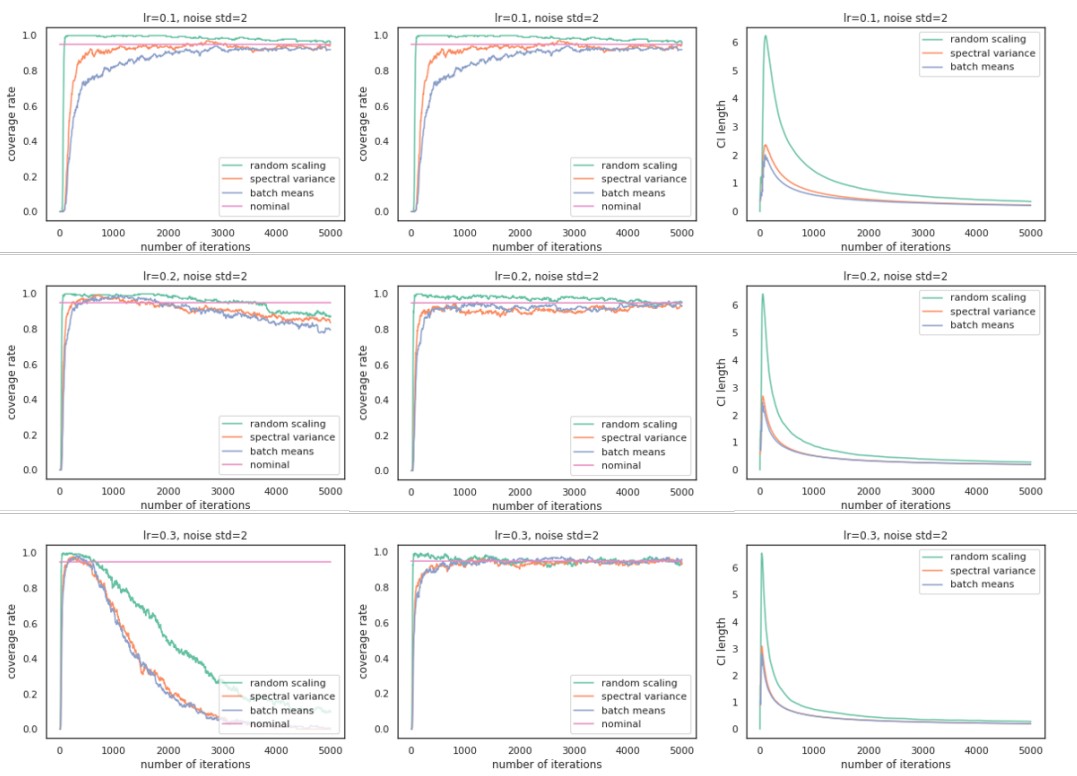

Figure 2: Q-Learning Results, $\sigma = 2$. Left: coverage rates for $Q^*$. Middle: coverage rates for $\mathbb{E}_{\mathcal{Q}_\eta} Q$. Right: confidence interval lengths.

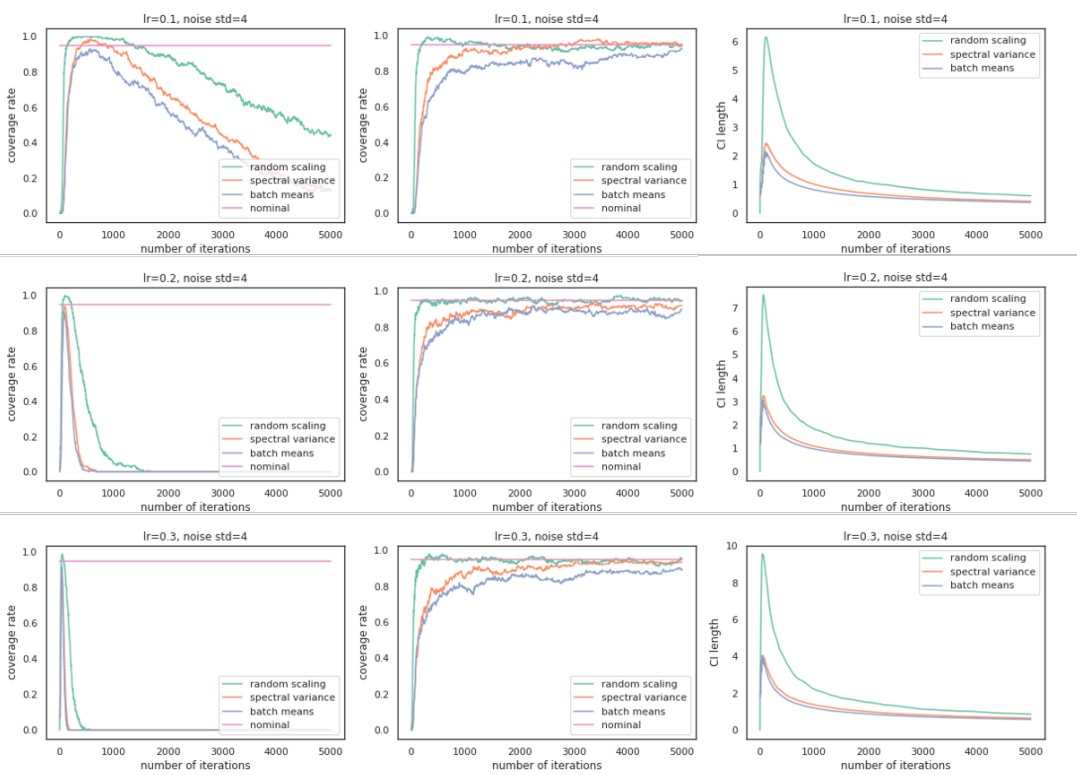

Figure 3: Q-Learning Results, $\sigma = 4$. Left: coverage rates for $\boldsymbol{Q}^*$. Middle: coverage rates for $\mathbb{E}_{\mathcal{Q}_\eta}\boldsymbol{Q}$. Right: confidence interval lengths.

Table 8: Q-Learning, $\sigma = 2$

| | | $n = 1,000$ | $n = 2,000$ | $n = 5,000$ |
|---|---|---|---|---|
| **$\eta = 0.1$** | | | | |
| Cov. for $\mathbb{E}_{\mathcal{Q}_\eta} \boldsymbol{Q}_1$ (%) | Spectral Variance | 92.0(1.918) | 93.5(1.743) | 95.0(1.541) |
| | Batch Means | 82.5(2.686) | 90.5(2.073) | 92.0(1.918) |
| | Random Scaling | 99.5(0.498) | 99.5(0.498) | 96.0(1.386) |
| Cov. for $\boldsymbol{Q}_1^*$ (%) | Spectral Variance | 92.0(1.918) | 93.5(1.743) | 95.0(1.541) |
| | Batch Means | 82.5(2.686) | 90.5(2.073) | 92.0(1.918) |
| | Random Scaling | 99.5(0.498) | 99.5(0.498) | 96.0(1.386) |
| Length | Spectral Variance | 0.707(0.076) | 0.430(0.045) | 0.231(0.024) |
| | Batch Means | 0.590(0.055) | 0.386(0.029) | 0.217(0.016) |
| | Random Scaling | 1.435(0.394) | 0.772(0.245) | 0.357(0.127) |
| **$\eta = 0.2$** | | | | |
| Cov. for $\mathbb{E}_{\pi_\eta} \boldsymbol{Q}_1$ (%) | Spectral Variance | 89.5(2.168) | 89.5(2.168) | 93.0(1.804) |
| | Batch Means | 93.5(1.743) | 92.5(1.862) | 93.5(1.743) |
| | Random Scaling | 97.5(1.104) | 98.5(0.859) | 95.5(1.466) |
| Cov. for $\boldsymbol{Q}_1^*$ (%) | Spectral Variance | 97.5(1.104) | 93.0(1.804) | 83.5(2.624) |
| | Batch Means | 98.0(0.990) | 94.5(1.612) | 79.5(2.854) |
| | Random Scaling | 99.0(0.703) | 98.0(0.990) | 87.0(2.378) |
| Length | Spectral Variance | 0.521(0.076) | 0.336(0.047) | 0.202(0.023) |
| | Batch Means | 0.511(0.052) | 0.333(0.032) | 0.199(0.015) |
| | Random Scaling | 0.877(0.304) | 0.524(0.195) | 0.282(0.118) |
| **$\eta = 0.3$** | | | | |
| Cov. for $\mathbb{E}_{\pi_\eta} \boldsymbol{Q}_1$ (%) | Spectral Variance | 91.5(1.971) | 91.5(1.971) | 93.5(1.743) |
| | Batch Means | 93.0(1.804) | 96.5(1.299) | 96.0(1.386) |
| | Random Scaling | 94.0(1.679) | 94.0(1.679) | 93.5(1.743) |
| Cov. for $\boldsymbol{Q}_1^*$ (%) | Spectral Variance | 69.5(3.255) | 23.0(2.975) | 0.0(0.000) |
| | Batch Means | 68.0(3.298) | 20.5(2.854) | 0.5(0.498) |
| | Random Scaling | 81.5(2.745) | 50.0(3.535) | 9.5(2.073) |
| Length | Spectral Variance | 0.491(0.072) | 0.330(0.043) | 0.201(0.023) |
| | Batch Means | 0.485(0.055) | 0.324(0.031) | 0.203(0.017) |
| | Random Scaling | 0.734(0.266) | 0.459(0.185) | 0.283(0.123) |

Table 9: Q-Learning, $\sigma = 4$

| | | $n = 1,000$ | $n = 2,000$ | $n = 5,000$ |
|---|---|---|---|---|
| **$\eta = 0.1$** | | | | |
| | Spectral Variance | 91.5(1.971) | 93.0(1.804) | 95.0(1.541) |
| Cov. for $\mathbb{E}_{\mathcal{Q}_\eta} \boldsymbol{Q}_1$ (%) | Batch Means | 79.5(2.854) | 85.0(2.524) | 93.0(1.804) |
| | Random Scaling | 96.0(1.386) | 94.5(1.612) | 95.0(1.541) |
| | Spectral Variance | 93.0(1.804) | 67.0(3.324) | 13.0(2.378) |
| Cov. for $\boldsymbol{Q}_1^*$ (%) | Batch Means | 81.0(2.773) | 57.5(3.495) | 14.0(2.453) |
| | Random Scaling | 98.5(0.859) | 89.5(2.168) | 44.0(3.509) |
| | Spectral Variance | 1.020(0.150) | 0.694(0.095) | 0.419(0.056) |
| Length | Batch Means | 0.821(0.100) | 0.588(0.060) | 0.382(0.032) |
| | Random Scaling | 1.726(0.648) | 1.072(0.414) | 0.612(0.237) |
| **$\eta = 0.2$** | | | | |
| | Spectral Variance | 87.0(2.378) | 89.5(2.168) | 92.0(1.918) |
| Cov. for $\mathbb{E}_{\pi_\eta} \boldsymbol{Q}_1$ (%) | Batch Means | 83.0(2.656) | 88.5(2.255) | 90.0(2.121) |
| | Random Scaling | 94.5(1.612) | 95.5(1.466) | 94.5(1.612) |
| | Spectral Variance | 0.0(0.000) | 0.0(0.000) | 0.0(0.000) |
| Cov. for $\boldsymbol{Q}_1^*$ (%) | Batch Means | 0.0(0.000) | 0.0(0.000) | 0.0(0.000) |
| | Random Scaling | 4.5(1.465) | 0.0(0.000) | 0.0(0.000) |
| | Spectral Variance | 1.093(0.205) | 0.779(0.126) | 0.506(0.067) |
| Length | Batch Means | 0.970(0.127) | 0.699(0.083) | 0.457(0.043) |
| | Random Scaling | 1.833(0.658) | 1.212(0.500) | 0.750(0.307) |
| **$\eta = 0.3$** | | | | |
| | Spectral Variance | 85.0(2.524) | 91.0(2.023) | 93.5(1.743) |
| Cov. for $\mathbb{E}_{\pi_\eta} \boldsymbol{Q}_1$ (%) | Batch Means | 79.5(2.854) | 86.0(2.453) | 89.0(2.212) |
| | Random Scaling | 94.5(1.612) | 94.0(1.679) | 95.0(1.541) |
| | Spectral Variance | 0.0(0.000) | 0.0(0.000) | 0.0(0.000) |
| Cov. for $\boldsymbol{Q}_1^*$ (%) | Batch Means | 0.0(0.000) | 0.0(0.000) | 0.0(0.000) |
| | Random Scaling | 0.0(0.000) | 0.0(0.000) | 0.0(0.000) |
| | Spectral Variance | 1.378(0.251) | 0.999(0.155) | 0.643(0.074) |
| Length | Batch Means | 1.202(0.157) | 0.877(0.097) | 0.578(0.049) |
| | Random Scaling | 2.229(0.923) | 1.513(0.586) | 0.861(0.341) |