# OpenReview forum: "A Statistical Online Inference Approach in Averaged Stochastic Approximation"
_NeurIPS.cc/2022/Conference — NeurIPS 2022 Accept_

### Official Review · Reviewer_2ACX · 2022-07-09

**Rating:** 7
**Confidence:** 5
**Soundness:** 4 excellent
**Presentation:** 3 good
**Contribution:** 4 excellent

**Summary:**

This paper considers the online statistical inference problem under a constant step size stochastic approximation setting. The authors first present a functional central limit theorem type result. Then given the FCLT result for the stochastic approximation problem, they further propose an online statistical inference framework using random scaling. They also illustrate their theoretical results in three cases, i.e., linear stochastic approximation, stochastic gradient descent, as well as Q-learning. Some numerical experiments are given to further support their findings.




**Questions:**

I have just one question for the authors: under the constant step SGD setting, the bias ( $\mathbb{E}_{\pi} \beta - \beta^*$) is not negligible, and in practice, the statistical inference cannot be done. The authors state that when $\eta \rightarrow 0$, the bias term is diminishing as well, but wouldn't that becomes the case where we have a diminishing step size and where traditional Polyak-Judisky type estimator and inference will be used instead?

**Strengths And Weaknesses:**

**Strengths**

- This paper is overall well written, and the authors present their theoretical findings clearly.
- In the literature, there is much work established on the diminishing step-size SGD and its corresponding statistical inference, from Polyak and Juditsky (1992). On the contrary, the constant step size SGD (and its parameter inference) is rarely studied, this paper enriches the theoretical results in this area.
- The authors also have a theoretical contribution to relaxing the $2+\delta$  moment condition.
- In Section 3, the authors also give theoretical results on three useful examples.

In summary, I would like to emphasize this paper's theoretical contribution to the constant step size stochastic approximation problem, both the asymptotic distribution part and the online statistical inference part.

**Weakness**

This paper is very good in general, I have a few concerns that are stated below,

1. Some related work should be cited properly, e.g.,  [1] present an online statistical inference framework for SGD-type estimator;
[2] gives the very first asymptotic results and a statistical inference framework for adaptively collected data; [3] gives some nice theoretical results on the constant step-size SGD.
2. The assumption (strong convexity) in Theorem 3.2 is a little bit strong, and cannot be applied to the logistic regression case that the authors used in the experiment section. The authors could find an extension to the locally strong convexity in [4], under which the logistic regression satisfies the assumption.

[1] Chen, Xi, Jason D. Lee, Xin T. Tong, and Yichen Zhang. "Statistical inference for model parameters in stochastic gradient descent." The Annals of Statistics 48, no. 1 (2020): 251-273.

[2] Chen, Haoyu, Wenbin Lu, and Rui Song. "Statistical inference for online decision making via stochastic gradient descent." Journal of the American Statistical Association 116, no. 534 (2021): 708-719.

[3] Dieuleveut, Aymeric, Alain Durmus, and Francis Bach. "Bridging the gap between constant step size stochastic gradient descent and Markov chains." The Annals of Statistics 48, no. 3 (2020): 1348-1382.

[4] Su, Weijie J., and Yuancheng Zhu. "Uncertainty quantification for online learning and stochastic approximation via hierarchical incremental gradient descent." arXiv preprint arXiv:1802.04876 (2018).

---

> ### Author Response · Authors · 2022-07-29
> **Response to Reviewer 2ACX**
>
> We greatly appreciate for your detailed review and suggestions. Here are our responses to your comments.
>
> "Some related work ... constant step-size SGD.": Thanks for pointing out potential related literature. This part will be added and polished in revision.
>
> "The assumption (strong convexity) in Theorem 3.2 is a little bit strong, ... satisfies the assumption.": The strongly-convex assumption is actually common in SA inference literature [2-4]. In our work, this ensures the Wasserstein contraction of SGD iterates and enables further application of our main Theorem 2.1. Locally strong convexity ensures the function to be strongly convex in any compact set $K\subset \mathbb{R}^d$, so an alternative solution may be appropriately choosing a (large) compact set, bounding the probability for iterates to escape the set, and using contraction arguments when iterates stay in the set (since strong convexity holds in $K$). Logistic regression falls in this case. We believe such extension is a meaningful direction for future research, which is beyond the scope of our current work.
>
> "Questions -- the case when $\eta\rightarrow 0$": I think there are still some differences between the case $\eta\rightarrow 0$ and Polyak-Judisky type estimator with diminishing step-size. In the latter, the step-size diminishes with number of  iterations; while in the former, the limits wrt number of iterations $n$ and step-size $\eta$ are independent and taken in order. The case $\eta\rightarrow 0$ is closer to continuous SDE approximation of SGD iterates, e.g., [1].
>
> [1] Li Q, Tai C, Weinan E. Stochastic modified equations and adaptive stochastic gradient algorithms[C]//International Conference on Machine Learning. PMLR, 2017: 2101-2110.
>
> [2] Polyak B T, Juditsky A B. Acceleration of stochastic approximation by averaging[J]. SIAM journal on control and optimization, 1992, 30(4): 838-855.
>
> [3] Chen X, Lee J D, Tong X T, Zhang Y. Statistical inference for model parameters in stochastic gradient descent[J]. The Annals of Statistics, 2020, 48(1): 251-273.
>
> [4] Lee S, Liao Y, Seo M H, Shin Y. Fast and robust online inference with stochastic gradient descent via random scaling[C]//Proceedings of the AAAI Conference on Artificial Intelligence. 2022, 36(7): 7381-7389.

---

> > ### Comment · Reviewer_2ACX · 2022-08-08
> > **Further Comments**
> >
> > I appreciate the responses from the authors. I have no further concern and would maintain my current rating.

---

### Official Review · Reviewer_KDyo · 2022-07-11

**Rating:** 4
**Confidence:** 5
**Soundness:** 2 fair
**Presentation:** 3 good
**Contribution:** 2 fair

**Summary:**

The paper suggests a procedure for constructing confidence intervals for constant step-size SA algorithms. The suggested procedure is based on FCLT and random scaling procedure. This technique might be viewed as an alternative to the methods based on estimations of the asymptotic variance of the Markov chain, such as batch mean or spectral variance. The method is tested on a constant step-size SGD procedure for low-dimensional linear and logistic regressions.

**Questions:**

1. I would be curious to see the empirical performance of the random scaling approach in RL applications. For example, I suggest the authors consider the grid-world type example and a Q-learning procedure on it. Another scenario could be TD-learning with linear functional approximation, studied e.g. in the paper [17] (cited in the paper);

**Limitations:**

The authors addressed the limitations correctly. Yet the main limitation is that only the constant step-size SA problems are considered, which lowers the potential applicability of the result.

**Strengths And Weaknesses:**

Originality.

Results similar to Theorem 2.1 are well-known in the Markov chain literature (see e.g. [Douc et al, 2018], sections 21.2-21.3). It is also typical that there are no additional requirements for the moments of order $(2+\delta)$. It is also possible to mention the paper [Atchade, 2013]. Generalizations of these results to the multidimensional scenario are rather straightforward. CLT for the Markov kernels which are W-uniformly geometrically ergodic (the scenario considered in Proposition 2.1) is also well-known in MCMC literature.
Moreover, CLT results in theorems 3.1--3.3 do not require novel methods and easily follow from obtaining the Wasserstein contractility of the corresponding transition kernels. So the theoretical findings of the paper are limited.  In the empirical evaluation, the authors also consider only model examples of low-dimensional logistic and linear regressions.

Clarity
The paper is quite well-written, and the suggested methodology and theoretical derivations are clear.  I would suggest to fix the statement of Theorem 3.3 in order to mention explicitly the access to generative model $P(\cdot | s,a)$ for each state-action pair $(s,a)$.

Significance.
Constant step-size SGD and $Q$-learning with constant step size are of moderate interest. Q-learning without bonuses and with constant step size makes sense only with access to generative model, which is of limited practical interest. More interesting scenarios invoke the decreasing step size in SA and/or the Markovian noise dynamics, which require much more sophisticated analysis. Another problem concerns the bias of constant SA scheme. For example, in theorem 3.2 the covariance matrix $\Sigma_{\pi_{\eta}}$ scales as $\|\|\Sigma_{\pi_{\eta}}\|\| \lesssim \eta$, which is of the same order as the bias term in th.3.2.2. Thus bounding the bias as $O(\eta^{1/2})$ is not enough.

References:
[Douc et al, 2018] - R. Douc, E. Moulines, P. Priouret, and P. Soulier. Markov chains. Springer Series in Operations Research and Financial Engineering, 2018.
[Atchade, 2013] Y.F. Atchad´e (2016), Markov Chain Monte Carlo confidence intervals. Bernoulli 22(3), 1808–1838.

---

> ### Author Response · Authors · 2022-07-29
> **Response to Reviewer KDyo**
>
> We greatly appreciate for your detailed review and suggestions. Here are our responses to your comments.
>
> "Results similar to Theorem 2.1 are well-known in the Markov chain literature.": We have been aware of the rich literature in Markov chains, and we have acknowledged that there are a bunch of CLT/convergence results [1-8] whose conclusions are similar to ours.  However, our motivation is to complement the stochastic approximation literature (more specifically, in the field of ML applications such as SGD and RL) with the performance of constant step-size algorithms: there are already works based on Polyak-Ruppert Averaging [10], most with decaying step-sizes and using customized decomposition of the stochastic noise; so what happens for the constant step-size SA algorithms? In such time-homogeneous case, could we adopt known Markov chain results to simplify and unify the proofs?
>
> To our knowledge, Theorem 2.1 is probably the only result that is suitable for SGD/RL applications. Most CLT results require the Markov chain to be either reversible or stationary [1,2,4,6] (SGD/RL iterates does not satisfy reversibility, and can start from arbitrary initial distributions), or satisfy unverifiable conditions (e.g., mixing conditions) [3,5-8], so these results do not work in our settings.
>
> "It is also typical ... order $(2+\delta)$.": In SA literature, our result removes this additional assumption in Theorem  2 of [11]. In addition, some works also require this technical condition (Assumption 3.2 of [12]) or even more stringent ones (e.g., Assumption 1(v) of [13], Assumption 3.2 of [14]), although they consider suitably decaying step-sizes. One purpose of our work is to make a meaningful comparison between such two different cases, and to pave the way for a possible further investigation: under what minimal assumptions (on step-sizes, etc.) can this $(2+\delta)$ condition be removed.
>
> "CLT for the Markov kernels which are W-uniformly geometrically ergodic ... in MCMC literature.": Our Proposition 2.1 is specifically oriented for SGD/RL applications. On the one hand, Proposition 2.1 allows for arbitrary $p, q\geq 1$. This is crucial in unifying the proofs, since for SGD with (traditionally) bounded 2nd moment noise, $p=q=2$ need to be used; while in TD/Q-learning, where the reward is assumed to be bounded and the Bellman operator is a contraction in $l_{\infty}$ sense, result at $p=\infty, q=1$ is required. On the other hand, we do not impose additional requirements on the transition kernel $P$ (e.g., reversibility), which may be preferable for future plug-in applications. Therefore, we believe Proposition 2.1 is useful and may be of independent interest.
>
> "Moreover, CLT results in theorems 3.1--3.3 ... transition kernels.": Although proved directly from geometric ergodicity, these are direct case studies used to illustrate the universality and applicability of Theorem 2.1 and Proposition 2.1. They also complement the SA literature with a series of CLT results with constant step-size algorithms, and may inspire further estimation and inference methods for parameters of such SA algorithms.
>
> "Q-learning without bonuses ... require much more sophisticated analysis.": Because current RL inference works mainly focus on Q-learning with a generative model [15-17],  we followed their settings. Nevertheless, we discuss our crude idea for the asynchronous case. In such case, only one component $(s_t,a_t)$ of the vector $Q_t$ is updated, and the multiplicative factor before $Q_t$ in (12) changes from $1-\eta$ to $I-\eta\Lambda_t$, where $\Lambda_t$ is a diagonal matrix with 1 at $(s_t,a_t)$-th diagonal and 0 otherwise. This multiplicative matrix factor is not positive-definite, so generally the Markov chain does not satisfy geometric ergodicity and cannot apply our results. Further assumptions may need to be considered. In addition, some analysis of decreasing step-size SGD/Q-learning already exists in literature [13,17].
>
> "I would be curious to see the empirical performance of the random scaling approach in RL applications.": We appreciate for your advice, and will consider RL experiments in its revision. We think the inference performance to be similar with SGD results: random scaling surpassing traditional methods in cov. rate accuracy while being conservative in CI lengths.
>
> Thanks again for pointing out potential related literature and writing parts that may raise confusion. These will be polished in revision.

---

> > ### Author Response · Authors · 2022-07-29
> > **Reference**
> >
> > [1] SEPPÄLÄINEN T. Notes on the martingale approach to central limit theorems for Markov chains[J]. 2006.
> >
> > [2] Maxwell M, Woodroofe M. Central limit theorems for additive functionals of Markov chains[J]. Annals of probability, 2000: 713-724.
> > [3] Jones G L. On the Markov chain central limit theorem[J]. Probability surveys, 2004, 1: 299-320.
> >
> > [4] Roberts G O, Rosenthal J S. General state space Markov chains and MCMC algorithms[J]. Probability surveys, 2004, 1: 20-71.
> >
> > [5] Utev S A. On the central limit theorem for φ-mixing arrays of random variables[J]. Theory of Probability & Its Applications, 1991, 35(1): 131-139.
> >
> > [6] Merlevède F, Peligrad M, Utev S. Functional CLT for martingale-like nonstationary dependent structures[J]. Bernoulli, 2019, 25(4B): 3203-3233.
> >
> > [7] Merlevède F, Peligrad M. Functional CLT for nonstationary strongly mixing processes[J]. Statistics & Probability Letters, 2020, 156: 108581.
> >
> > [8] HAFOUTA Y. Convergence rates in the functional clt for α-mixing triangular arrays[J]. arXiv preprint arXiv:2107.02234, 2021.
> >
> > [9] Derriennic Y, Lin M. The central limit theorem for Markov chains started at a point[J]. Probability theory and related fields, 2003, 125(1): 73-76.
> >
> > [10] Polyak B T, Juditsky A B. Acceleration of stochastic approximation by averaging[J]. SIAM journal on control and optimization, 1992, 30(4): 838-855.
> >
> > [11] Mou W, Li C J, Wainwright M J, Bartlett P L, Jordan M I. On linear stochastic approximation: Fine-grained Polyak-Ruppert and non-asymptotic concentration[C]//Conference on Learning Theory. PMLR, 2020: 2947-2997.
> >
> > [12] Li X, Liang J, Chang X, Zhang Z. Statistical Estimation and Online Inference via Local SGD[C]//Conference on Learning Theory. PMLR, 2022: 1613-1661.
> >
> > [13] Lee S, Liao Y, Seo M H, Shin Y. Fast and robust online inference with stochastic gradient descent via random scaling[C]//Proceedings of the AAAI Conference on Artificial Intelligence. 2022, 36(7): 7381-7389.
> >
> > [14] Chen X, Lee J D, Tong X T, Zhang Y. Statistical inference for model parameters in stochastic gradient descent[J]. The Annals of Statistics, 2020, 48(1): 251-273.
> >
> > [15] Hao B, Ji X, Duan Y, Lu H, Szepesvari C, Wang M. Bootstrapping Fitted Q-Evaluation for Off-Policy Inference[C]//International Conference on Machine Learning. PMLR, 2021: 4074-4084.
> >
> > [16] Khamaru K, Xia E, Wainwright M J, Jordan M I. Instance-Dependent Confidence and Early Stopping for Reinforcement Learning[J]. arXiv preprint arXiv:2201.08536, 2022.
> >
> > [17] Li X, Yang W, Zhang Z, Jordan M I. Polyak-Ruppert Averaged Q-Leaning is Statistically Efficient[J]. arXiv preprint arXiv:2112.14582, 2021.

---

> > > ### Comment · Reviewer_KDyo · 2022-08-04
> > > **From stationary to non-stationary**
> > >
> > > - Theorem 2.1 is, of course, "almost" an "if and only if" condition to obtain an FCLT [the condition is very similar to that of Maxwell and Woodroofe].
> > > - For Markov chains, it is generally straightforward to derive a non-stationary result from a stationary result, see Douc et al, "Markov chains", Chapter 21 - Section 21.1.2 explains the path from stationary to non-stationary under the assumption of phi-irreducibility. Section 21.4.2 describes a different approach applicable to non-irreducible kernels (satisfying ergodicity property in generalized Wasserstein distance): the idea there is based on the coupling of non-stationary and stationary version of the chain, to establish that the difference on these two versions in o_P(1) [there are many ways to do this]. So there are at least two approaches which are readily applicable to go from stationary to non-stationary

---

> > ### Author Response · Authors · 2022-08-04
> > **Q-Learning Experiment Added**
> >
> > We have added a Q-learning experiment on Grid World in our rebuttal revision (detailed in Appendix D). The overall performance is similar to that of SGD: random scaling produces wider CI lengths while being faster to achieve the coverage rate accuracy than other methods (spectral variance & batch means). As for TD-learning with linear function approximation, since it is a linear stochastic approximation (LSA) problem, we think its performance is likely to highly resemble that of SGD for linear regression.

---

### Official Review · Reviewer_hS9V · 2022-07-11

**Rating:** 6
**Confidence:** 3
**Soundness:** 4 excellent
**Presentation:** 4 excellent
**Contribution:** 3 good

**Summary:**

This paper considers a statistical inference problem for which parameters are estimated by stochastic approximation (SA) with a constant step size. The main theoretical results are derivations of functional central limit theorems (FCLTs) under a Markov chain setting. Three important cases are analyzed: linear stochastic approximation (LSA), stochastic gradient descent (SGD), and Q-learning. The inference method is based on random scaling that has been recently proposed in the literature. The numerical experiments involve SGD in the context of linear and logistic regression with a large sample but a small number of covariates.

**Questions:**

(1) The nonzero bias problem is unfortunate, as the confidence interval does not center around the true parameter value. It is nice to have an upper bound on the bias at the order of the square root of the learning rate.  Is there any way to develop lower bounds on the bias? It would be interesting to know what type of nonlinearity matters for the bias.

Here are some minor suggestions.

(a) References 13 and 14 are now in the conference proceedings, 13 in AAAI-22 and 14 in COLT-22, respectively.

(b) In line 205, what is "i" in collected i the $t$-th iteration; perhaps "in"?





**Limitations:**

Generally speaking, the authors addressed the limitations. A few things, which can be mentioned (additionally or in more detail), are as follows:

(1) SGD with a constant step size leads to an inefficient estimator and as a result, the resulting confidence interval with random scaling will be wider than that with a decreasing step size.

(2) How to choose the learning rate (step size) in applications is not discussed; furthermore, there is no real data application which would provide a more difficult environment for choosing the step size.

(3) No replication files are provided in the supplementary material.

**Strengths And Weaknesses:**

- Originality: Recent work has applied random scaling to SGD with a decreasing step size for the purpose of inference. This paper is new in the sense that it applies random scaling to SGD with a constant step size.

- Quality: The theoretical work is of high quality. Assumptions are clearly stated and the results are impressive. The numerical work is of medium quality. The experimental design is simple but adequate to make the point that random scaling works for SGD with a constant step size in the linear regression model.

- Clarity: The paper is very clear.

- Significance: The results in the paper are non-trivial and useful. The inference method is incomplete when the bias is nonzero but the results in this paper may lead to a new line of research that can estimate or bound the bias.

---

> ### Author Response · Authors · 2022-07-29
> **Response to Reviewer hS9V**
>
> We greatly appreciate your effort and constructive instructions. Here are our responses to your comments.
>
> "Is there any way to develop lower bounds on the bias?": We believe analyzing such lower bounds is a promising direction for future research, since to carry out inference for $\beta^*$, it is necessary to characterize the magnitude of this bias. Bounds on such bias may vary according to specific settings, and currently such analysis is rare to our knowledge.
>
> "How to choose the learning rate ...": As shown in our theory and experiments, generally, constant step-size SGD requires smaller step-sizes to ensure convergence, as compared to diminishing step-size algorithms. This may serve as a practical guidance in applications.
>
> Thanks again for pointing out typos and citation issues. These will be fixed in revision.

---

> > ### Comment · Reviewer_hS9V · 2022-08-08
> > **Further comments**
> >
> > I appreciate the responses from the author(s). One minor point is that as commented originally, no replication files are provided in the supplementary material. Would the author(s) provide all replication files in the camera ready version if this paper is accepted?

---

> > > ### Author Response · Authors · 2022-08-08
> > > **Further response**
> > >
> > > Yes. All replication code for experiments will be uploaded in the camera ready version (in case there may be additional experiments before that) if accepted.

---

> > > > ### Comment · Reviewer_hS9V · 2022-08-09
> > > > **Thanks for the response**
> > > >
> > > > OK. Thank you.

---

### Meta-Review · Area_Chair_nocs · 2022-08-23

**Recommendation:** Accept
**Confidence:** Less certain

**Metareview:**

This paper provides an online framework for stochastic approximation algorithms with fixed step size. Several reviewers commended the work, and I agree that the contributions are sound and relevant to the NeurIPS community. The more negative review provides very constructive criticism, and the knowledgeable reviewer points out flaws that should be addressed in the final version. The authors have done a good job responding to reviewer comments, and have added additional experimental details to bolster the results. Some discussion on the limitations of restricting to constant step size regimes (including some of the points laid out in the discussion/responses to reviewers) will also help readers contextualize and clarify the scope

**Award:**

No

---

### Decision · Program_Chairs · 2022-09-14

Accept